# Uncovering the phonon spectra and lattice dynamics of plastically deformable InSe van der Waals crystals

Jiangtao Wu[1,13], Yifei Lin[2,13], Mingfang Shu[1,13], Yifei Liu[3], Yupeng Ma[3], Gaoting Lin [1], Cuiping Zhang[1], Pengfei Jiao[1], Fengfeng Zhu [4], Yan Wu [5], Russell A. Ewings [6], Helen C. Walker [6], Guochu Deng[7], Songxue Chi [5], Shengwei Jiang [1], Matteo Baggioli [8], Min Jin[9], Haozhe Wang [10], Weiwei Xie [10], Tian-Ran Wei [3] ✉, Jiong Yang [2] ✉, Xun Shi [3,11] & Jie Ma [1,12] ✉

Stacking two-dimensional (2D) van der Waals (vdW) materials in a layered bulk structure provides an appealing platform for the emergence of exotic physical properties. As a vdW crystal with exceptional plasticity, InSe offers the opportunity to explore various effects arising from the coupling of its peculiar mechanical behaviors and other physical properties. Here, we employ neutron scattering techniques to investigate the correlations of plastic interlayer slip, lattice anharmonicity, and thermal transport in InSe crystals. Not only are the interlayer slip direction and magnitude well captured by shifts in the Bragg reflections, but we also observe a deviation from the expected Debye behaviour in the heat capacity and lattice thermal conductivity. Combining the experimental data with first-principles calculations, we tentatively attribute the observed evidence of strong phonon-phonon interactions to a combination of a large acoustic-optical frequency resonance and a nesting effect. These findings correlate the macroscopic plastic slip and the microscopic lattice dynamics, providing insights into the mechano-thermo coupling and modulation in 2D vdW materials.

As crystalline solids composed of atomically thin layers, two-dimensional (2D) materials provide a promising opportunity of engineering the individual layers separately and then stacking them together by manipulating the van der Waals (vdW) forces. Thereafter, consecutive layers can demonstrate different orientations and thus a series of exotic electronic, optical, thermal, and mechanical properties could be built up for large-size and single-domain productions in a controlled manner[1]. If the layers are stacked thickly, the 2D vdW crystals could be developed with the bulk-like transport properties and could be directly applied in advanced energy batteries, thermoelectrics, and optoelectronics.

Recently, exceptional plastic deformability was discovered in 2D vdW crystal InSe[2,3], a IIIA-VIA semiconductor that has already been found to show rich and tunable physical properties[4–6]. The crystals exhibit noticeable compression strains around 80%, and can be morphed into various shapes without breaking into pieces[2]. Unlike the commonly seen flexibility induced by small modulus and low thickness, the plasticity is proposed to arise from both the interlayer slip (relative glide of the adjacent basal planes) and the cross-layer slip (slip of non-(001) planes). As the dominant plastic deformation mechanism for layered crystals, several possible slip paths have been suggested for this interlayer slip[7]. Undoubtedly, the plastic interlayer slip should strongly mediate the structure and thus multifarious physical properties, offering more perspectives to understand and modulate these 2D vdW crystals.

**Fig. 1 | Crystal structure and elastic diffractions of β-InSe. a** Crystal structure of β-InSe and the projection plane of (001) and (100); **b** Selected area electron diffraction (SAED) pattern of (001) plane; **c** Reciprocal space map of (HK0) reflection plane ($-1 \leq E \leq 1$ meV) from neutron scattering measurements, performed using the MERLIN spectrometer at ISIS. The white hexagons in **c** denote the Brillouin Zones. The concentric rings in the MERLIN data are from diffraction by the aluminum

sample holder. The yellow arrows in **b** and **c** are the distance of the nearest diffraction points to the Brillouin zone center by electrons and neutrons, respectively; **d** Three-dimensional pattern of the diffuse spindle geometry ($-1 \leq E \leq 1$ meV) at 200 K, MERLIN. Inset schematically shows the interlayer slip in real space. The diffuse scattering intensities along the [001] direction at **e** (K-KL) plane and **f** (HHL) plane, respectively.

However, the understanding on such a seemingly simple slip is quite inadequate in several aspects. Firstly, there is currently no direct experimental verification and identification of the slip path despite several theoretical predictions[7,8]. In fact, it is already challenging even to determine the crystal structure of 2D vdW crystals, since they can exhibit different prototypes (2H, 3 R, etc.) along the abundant stacking faults induced by the slip[7]. Most probing methods (such as TEM) are powerful as to local regions[9–13], yet fail to give an analysis of the whole crystal. Secondly, the underlying physical origin of this interlayer slip needs to be clarified. It has been recognized that the interlayer slip barrier energy is quite low for the vdW materials due to the weak interlayer interactions. Nevertheless, a complete understanding of the correlation between the slip and lattice dynamics is missing, i.e., large atomic displacement should be closely related to the local lattice vibrations and the phonon dispersions. Thirdly, the effects of such a slip, in turn, on the lattice dynamics and transport properties remain unexplored. The interlayer slip causes dense stacking faults and structural disorder across the vdW gap, which will largely disrupt the periodicity along the *c*-axis and certainly affect the transport properties.

Here, taking the plastically deformable InSe vdW crystal as a case study, we elaborate the correlation between interlayer plastic slip, lattice anharmonicity, and thermal transport in 2D vdW crystals by combining neutron scattering experiments with the theoretical analyses. Experimentally, the interlayer slip is verified, and the slip path is identified. Such a slip is further found to be closely related to the instability of the soft optical shear mode with very low energies. In turn, the slip further amplifies the phonon anharmonicity, not only broadening the phonon dispersion (corresponding to a large phonon scattering rate), but also causing one strongly damped acoustic-mode, a phenomenon exclusive to highly disordered or even liquid-like materials. Unlike augmenting the three-phonon scattering channels SnTe/PbTe mainly around the zone center, the local phonons of InSe present "nesting" behavior with two paralleled phonon groups over a large q-range and enable more acoustic-optical three-phonon scattering channels which amplifies the anharmonicity[14]. These factors

together bring about important deviations of the low-temperature heat capacity from the Debye model and enhanced phonon scattering, leading to low thermal conductivity. This work provides a direct insight into the mechano-thermo coupling of 2D vdW crystals and expands the realm of stacking a layered material towards a bulk crystal.

## Results and discussion
### Crystal structure and interlayer slip
High-quality InSe crystals are prepared by the Bridgeman method[5] (see Method for detail). Figure 1b and c exhibit the diffraction patterns by electron scattering and neutron scattering, respectively. Clearly, a hexagonal symmetry is observed in the selected area electron diffraction (SAED), and the distance of nearest diffraction points of electrons, $\Delta q_{SAED}$, is ~2.98 nm$^{-1}$, which indicates that the material is in the 2H (hexagonal structure with an AB stacking) rather than the 3 R phase (rhombohedral structure with an ABC stacking, for which a spacing of reciprocal lattice points of ~5 nm$^{-1}$ is expected). We further employed elastic neutron scattering to capture the overall information of the bulk crystals by virtue of the zero charge of neutrons and thus the large penetration into the specimen without slicing the crystal to induce extra strain or phase transition[15]. As shown in Fig. 1c, the $q$ distance of two nearest Brillouin zone center in the elastic neutron scattering plane, $\Delta q_{INS}$ is 1.88 Å$^{-1}$, which well follows the $\Delta q_{INS} = 2\pi\Delta q_{SAED}$ relation. The observed reflection condition, L = 2n (a period of 2) for (−1−1L) reflections is in agreement with the 2H rather than 3 R symmetry as shown in Fig. 1e (data in black). There are two kinds of 2H structures, i.e., the β- (P6$_3$/mmc, with inversion center) and the ε-InSe (P-6m2, without inversion center). As shown in Fig. 1d–f, the Bragg reflections in (HHL) and (K-KL) plane comply the condition with L = 2n instead of L = 2n + 1, which point towards the β- rather than the ε-phase. Moreover, the single crystal X-ray diffraction has been conducted, and the refinement results indicate that the sample are probably the 2H β-phase rather than the 2H ε-phase, Supplementary Fig. 1. In addition, the Raman spectra, Supplementary Fig. 2, shows that there is no peak around 199 cm$^{-1}$, which is also consistent with the feature of the β-InSe[13].

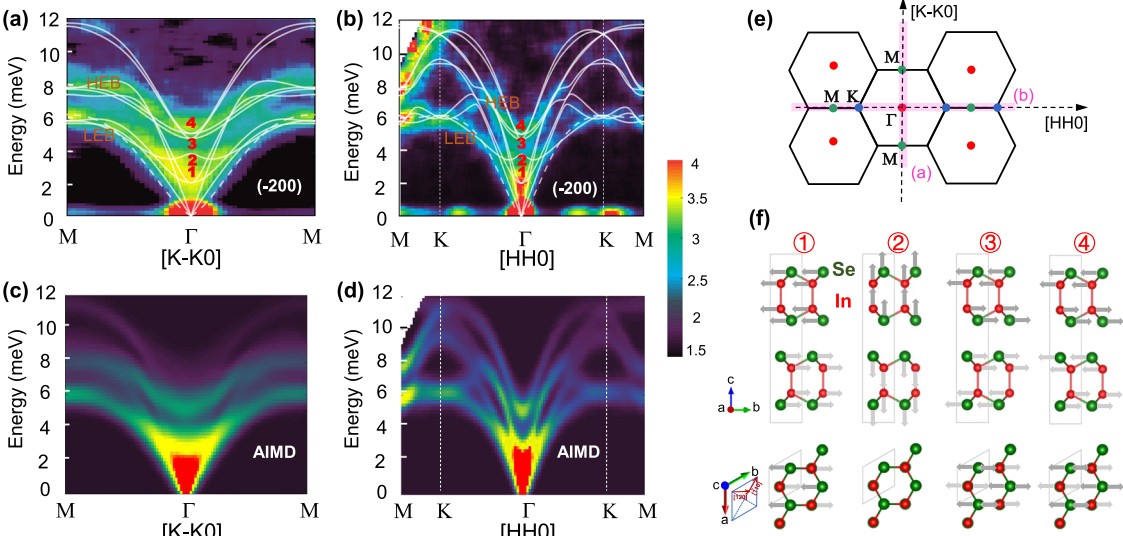

**Fig. 2 | Phonon dispersions, molecular dynamics simulations, and the related vibrating modes.** The phonon dispersions observed in the dynamical susceptibility $\chi''(q, E)$ from **a**, **b** inelastic neutron scattering (INS) measurements and **c**, **d** ab initio molecular dynamics (AIMD) simulations along [K-K0] and [HH0] directions at 200 K in the (-200) Brillouin zone, respectively; the white lines in **a** and **b** are the AIMD-calculated phonon dispersions and the dash-lines are highlighted for the ZA phonon dispersions; **c** and **d** are intensity plots considering the calculated dynamical structure factor from AIMD and instrument resolution. The $q$ represents the momentum transfer, E is energy, LEB and HEB denote low energy band and high energy band, respectively. **e** Schematics of the Brillouin zones with the high symmetry points indicated (Γ, K, M; the high symmetry directions for the mappings in **a** and **b** are marked by pink shade, respectively. **f** Illustration of the four optical phonon modes at Γ, as labeled in **a** and **b**, the arrows represent the vibrational directions of In and Se atoms in different modes.

More subtle structural information can be further obtained from the diffuse scattering signals, such as static and dynamic atomic displacements and extended lattice defects. Figure 1d shows the fairly diffusive, non-confined signals along c-axis, and the vertical spindle-like diffuse signal between Bragg peaks extends across all the measured Brillouin zone. This is different from what one would get from an isotropic single crystal, for which the signal would be composed of spheres in reciprocal space of limited radius. Here, extended diffuse scattering is observed along the c-direction while circle-like domains remain in the ab-plane. The continuous diffuse signal indicates the short-range ordering along c-axis and is ascribed to the weak interlayer vdW interaction of this bulk crystal. The absence of long-range order along the c-axis suggests the possible emergence of largely disordered dynamics within the 3D crystal and large non-elastic displacements in that direction, which is likely to modulate the phonon transport.

The interlayer slip path and displacement can be identified based on the diffuse scattering signals. As shown in Fig. 1e, there is an obvious shift of the reflection peaks for the (K-KL) planes with adjacent series of K value. By contrast, there is no shift for the reflections at (HHL) planes (Fig. 1f). This observation clearly indicates that there is a relative glide (or slip) of the adjacent (001) planes that tends to happen along the [1−10] direction rather than along the [110] direction. The shift value is about 0.75 rlu with reduced unit along [00L], corresponding to the slip displacement of 2.58 Å in real space (see Supplementary Fig. 3 for geometry and calculation details). The experimentally observed direction and displacement of the interlayer slip are consistent with previous calculations of the generalized stacking fault energy (GSFE)[7,8]: the energy barrier for the slip along the [1−10] direction (specifically, $1/3[120] + 1/3[\bar{1}10]$) is much lower than the [110] direction, and the displacement is predicted to be around 2.30 Å. Furthermore, the slipped structure corresponding to the slip vector (1/3, 2/3, 0) shows a comparatively low energy compared to the unslipped structure; the energy difference is nearly zero according to previous calculations[3,16]. This means such a slipped structure is quite stable and explains well the dense stacking faults in InSe vdW crystals. Although we cannot exclude other defects like dislocations, ripplocations, and point defects, the interlayer-slip-induced stacking faults should be the dominant.

## Phonon spectra and lattice dynamics

To investigate the slip effect on the lattice dynamics, the inelastic neutron scattering (INS) technique was applied to obtain the phonon dispersions of InSe (see SI for measurement details). AIMD simulations at 200 K for β-InSe were run and the results for the phonon dispersion (without linewidth) are overplotted as white lines, Fig. 2a and b. AIMD-simulated $\chi''(q, E)$ slices (including linewidth) along [K-K0] and [HH0] directions are plotted in Fig. 2c, d. Compared with the experimental INS data at the same temperature and Brillouin zone in Fig. 2a, b, the features of the spectrum can be captured by AIMD including phonon energies and INS intensities. However, the observed phonon dispersions deviate from the AIMD calculations for the low-energy phonons. Particularly, the out-of-plane transverse acoustic branch (denoted as ZA with dash-lines, and similar to that in 2D materials) with the lowest energy seems to be strongly damped in the experimental data. The damping nature of this phonon mode, vibrating along c-axis and propagating within the layers, is likely due to the dense disorder across the vdW gaps induced by the interlayer slip. More evidence for this possibly overdamped mode will be provided later.

Meanwhile, dense, low-energy optical modes cross over with each other and are strongly correlated with acoustic modes in the whole Brillouin Zone, particularly, the modes below 5 meV. The pronounced phonon energy/frequency overlap induces the intensive resonant scattering or frequency resonance effect[17,18]. As it is hard to explicitly distinguish these specific branches, two phonon bands are grouped as low energy band (LEB) and high energy band (HEB). The LEB is mixed with both acoustic and optical phonon modes, while the HEB are composed of only optical phonons. At the zone center (Γ-point), the TO mode of LEB corresponds to shear vibration, or oscillated sliding between the layers (see mode "1", Fig. 2f), sharing the same vibrational direction as of the [1-10] slip direction. This mode can easily interact strongly with the acoustic phonons due to the proximity of their energies, contributing to the large phonon-phonon scattering and enhancement of the anharmonicity. Unlike the 2D-like scattering of SnSe[19,20] and nano-domain effect of AgSbTe$_2$[21], InSe behaves, to some extent, as the "waterfall" effect in PbTe[22] and "nesting" effect in SnTe[14]: the excitation mode "1" TO

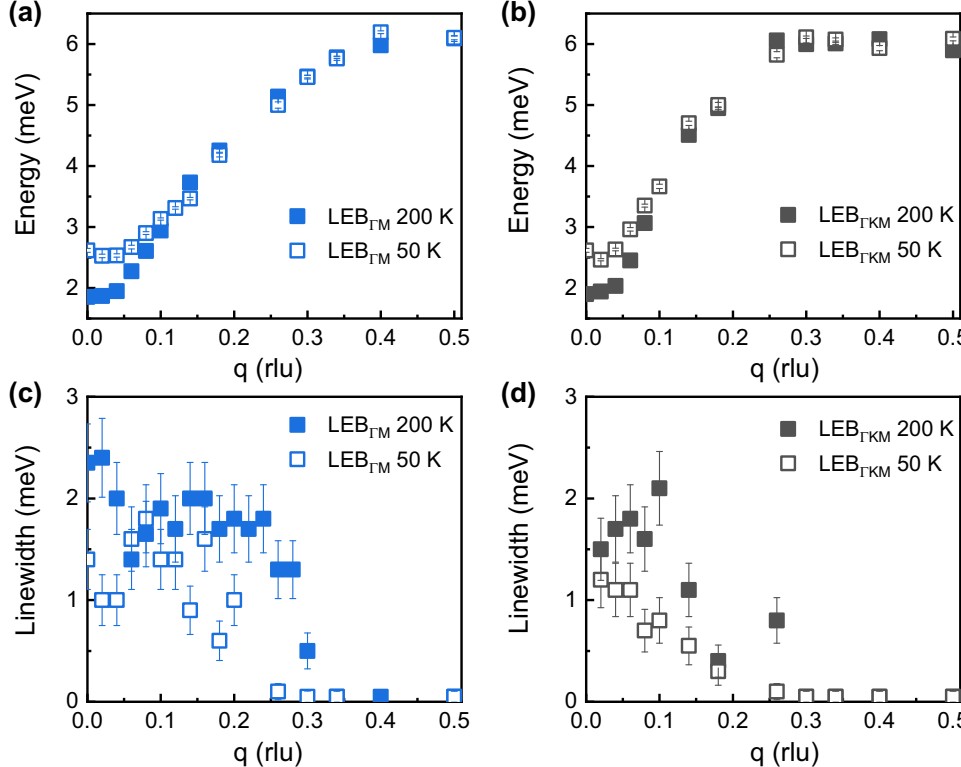

**Fig. 3 | The fitted phonon dispersions and linewidths along ΓM and ΓKM directions based on the experimental INS data. a, b** The comparison of *q* dependent phonon dispersions along ΓM and ΓKM directions at 50 K and 200 K, respectively; **c, d** The *q* dependent phonon linewidth along ΓM and ΓKM directions at 50 K and 200 K, respectively. The data and corresponding error bar are obtained from the fitting of constant-*q* cuts through INS data, rlu is the abbreviation for the relative unit.

mode at Γ (*q* = 0) is broad, even down to 0 meV, which is ascribed to the weak vdW force between the layers and the strong TA-TO interaction. Due to the low excitation energy (~2 meV) and prominent scattering intensity of the shear mode, the interlayer slip should be easily induced even at very low thermal excitation. This is consistent with the low slip barrier energy (~20 mJ/m² in refs. 7,8.]) as 17.6 meV per unit cell (2.2 meV per atom). Above the TO mode, there is one LO breath mode between adjacent layers, at around 3.8 meV (mode "2", Fig. 2f). The purely optical HEB reflects two vibration modes associated with the intralayer shear as shown in "3" and "4" of Fig. 2f, respectively. Interestingly, the HEB and LEB bands are roughly parallel to each other along ΓM direction, which is known as "nesting"[14]. This is a sign of numerous three-phonon scattering channels for the TO modes, significantly enhancing the intensity of acoustic-optical phonon interaction. Due to the strong TA-TO interaction, the phonon linewidths of LEB are very broad and the corresponding short phonon lifetimes are suggested by anharmonicity. Moreover, the nesting effect seems more prominent along the ΓM([K-K0]) direction than the ΓKM([HH0]) direction, which echoes the interlayer slip direction as discussed above.

Figure 3a and b show LEB dispersions along ΓM ([K-K0]) and ΓKM ([HH0]) directions at 50 K and 200 K, respectively. The data are obtained from the fitting of constant-q cuts through INS data. The phonon energy of LEB at the Γ-point, decreases from 2.7 meV at 50 K to 1.9 meV at 200 K. A similar significant softening is also obtained by AIMD simulations as shown in Supplementary Fig. 4d and Raman spectra in Supplementary Fig. 4f. Our AIMD results indicate that the frequency of the LEB mode decreases from 3.41 meV (50 K) to 2.03 meV (200 K) and 1.92 meV (400 K). Such an apparent temperature-dependent softening probably originates from the nesting acoustic-optical mode contribution of LEB and substantiates the phonon instability, which would further promote the interlayer slip. Meanwhile, an anomaly of the phonon

width is observed for InSe crystals. Unlike the regular phonon with increasing linewidths from the BZ zone-center to boundaries[21,22], the linewidths of LEB have larger values at zone-center for InSe. For small *q* regions, the optical phonon modes, especially LEB mode "1", are quite low and can strongly amplify the linewidth via acoustic-optical mode interactions. The phonon linewidths, $2\Gamma_j(q)$, which are inversely related to the phonon lifetimes via $\tau_j(q) \sim 2\pi / \Gamma_j(q)$, are extracted. (Constant-q cuts used for linewidths fitting, and the energies and linewidths of HEB are shown in Supplementary Figs. 5 and 6, respectively.) All phonon modes are broad (1 ~ 3 meV) comparable to the typical binary thermoelectric alloys with the strong phonon-lattice or phonon-phonon interaction as PbTe and SnTe[14,23]. The phonon linewidth (at Γ-point) of PbTe (incipient ferroelectric phonon-lattice interaction) increases with increasing temperature[22], while a decrease of the TO linewidths is observed with increasing q for SnTe[14,23]. Since the phonon linewidths of InSe have the similar temperature- and q-behaviors, the phonons are also considered to be strongly damped with large anharmonicity.

## Heat capacity

Figure 4a presents the measured heat capacity, $C_p$, and the calculated results by incorporating the AIMD (200 K) derived phonon density-of-states (DOSs), and the lattice dilation (the thermal expansion effect from the anharmonic phonons, $C_D$) (see SI for details). The experimental and computational data are consistent with each other between 50 K and 200 K, while the calculated $C_p$ is larger than the measured one below 50 K, inset of Fig. 4a. This deviation from the Debye model suggests strongly damped phonons even at low temperature and a clear difference from the typical behavior of 3D crystals[24]. As shown in Fig. 4b, the deviation is more clearly demonstrated by an obvious hump in the $C_p/T^3$ vs. *T* plot at around 12 K, and the calculated $C_{AIMD-tot}/T^3$ is considerably higher than the experimental one.

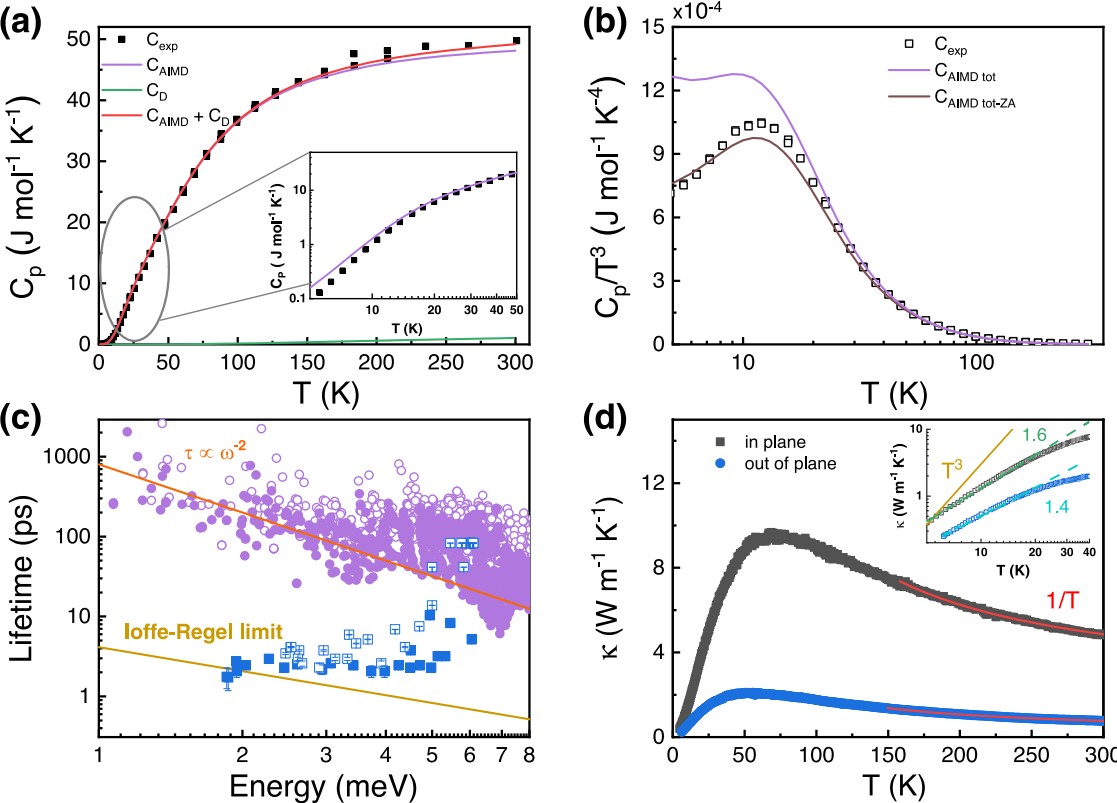

**Fig. 4 | The heat capacity, phonon lifetime, and thermal conductivity by experiments and simulations. a** The measured specific heat $C_p$ (black squares) and simulated specific heat $C_{AIMD}$ (purple curve) from the calculated phonon DOS, and the lattice dilation (the thermal expansion effect from the anharmonic phonons, $C_D$). Inset is the enlarged region from 0 K to 50 K. **b** The $C_p/T^3$ of InSe from measurement (hollow black squares), and the calculations according to the total phonon DOS (purple curve) and the partial phonon DOS without the ZA contribution (brown curve). DOS denotes phonon density-of-state, ZA is the out-of-plane transverse acoustic branch, and $T$ is temperature; **c** The temperature-dependence of energy vs. phonon lifetimes (blue solid squares for 200 K, blue hollow squares for 50 K) and the simulation (purple solid circles for 200 K, purple hollow circles for 50 K). The yellow solid line represents the Ioffe Regel limit, while $\omega$ is the phonon frequence and the orange one is guidance for the tendency of calculating lifetime. The lifetimes increase with phonon energies. **d** In-plane (black squares) and out-of-plane (blue circles) thermal conductivity. The inset shows the low temperature thermal conductivity. The red line is the fitting result of the $T^{-1}$ law above 150 K.

Although similar behavior could be observed in a large plethora of crystalline materials such as thermoelectrics[25–27], incommensurate structures[28,29], superionic conductors[30], orientationally disordered crystals[31], ferroelastic memory alloys[32], metal halide perovskites[33], ferroelectric materials[34], organic materials and even molecular crystals without any clear sign of disorder[35], the fundamental origin of this phenomenon is still an open question with wide interest and strong debate[36–42]. The thermal properties in these crystalline materials are evidently not a direct manifestation of positional disorder[43–45], for example glasses[46,47], but related to a phonon softening or to strong phonon-phonon, often as a result of enhanced anharmonicity[30,48–52]. As of now, the universal character of these Boson-peak-like features is still under scrutiny[53]. The deviation from the Debye behaviour in crystalline or weakly-disordered systems seems to be a concrete possibility, hinting towards possible different origins. We will discuss the case of InSe within this general picture.

To better understand such a deviation, the ZA contribution is tentatively subtracted from the total phonon DOS, $C_{AIMD-tot-ZA}/T^3$ to re-simulate the data. As shown in Fig. 4b, the calculated results agree with the measurement when the ZA contribution is removed. This is consistent with the results from INS in Fig. 2 and possibly the damping nature of the ZA mode. In Supplementary Fig. 7, the Debye-Einstein model was applied to fit the partial-phonon contributions, and the Boson-peak-like signal is mainly attributed to a low-energy Einstein mode ($Ei_I$) with $\theta_{Ei_I}$ 55.2 K (4.75 meV), which demonstrates the localized nature of the low-energy optical modes[54]. Moreover, the contribution of each phonon branch to the partial heat capacity ($C_p/T^3$) was presented in Supplementary Fig. 8.

From a microscopic perspective, the large anharmonicity should lead to the piling up of soft optical modes and for the overdamped-like nature of the ZA mode. That is, the combination of strong acoustic-optical phonon interaction and the phonon nesting dispersion signature should largely enlarge the three-phonon scattering space along [K-K0]. According to the origin of ZA and ZO, the ZA mode is the transverse acoustic branch along the slip surface and it is hard to build up the atomic long-range vibrations between slipping layers, while the ions vibrate toward each other for the ZO modes, which shorten the distance of the ions on different layers and could still organize the coherent vibration.

### Phonon lifetime and thermal conductivity

Figure 4c shows both energy- and temperature-dependence of phonon lifetime from INS experiment ($\tau_{exp}$) and the calculation by AIMD, both at 50 K and 200 K. Although the $\tau_{AIMD}$ follows the $\omega^{-2}$ scaling as a sign of propagating phonon modes, the measured $\tau_{exp}$ is orders of magnitude lower than the calculated one, and shows a weak dependence on frequency (energy), which is common for disordered or amorphous materials[55–57]. This discrepancy is another signature of the strong deviations from harmonic dynamics as assumed in the AIMD calculations. The large disorder of InSe crystals matches well with the significantly diffuse scattering signals. The structural origin of this disorder probably lies in the interlayer slip and the consequential

stacking faults. The Ioffe-Regel (IR) limit, $\tau_{IR} = 2\pi/\omega$,[58] where the normal vibration modes become attenuated and their mean free path gets comparable to the wavelength, is also plotted. As shown in Fig. 4c, the $\tau_{exp}$ falls below the IR limit for the low-energy phonons but exceeds this limit for high-energy phonons, corroborating the large anharmonicity of the phonons near the BZ center and the consequent disappearance of coherent and propagating quasiparticles at low wavevectors. Around 2 meV, the phonon lifetimes are shorter than or at least comparable to the IR limit, suggesting that the corresponding modes are not typically propagating but rather of quasi-localized diffusive nature[58]. Increasing the energy, the lifetimes are approximately constant but still reasonably close to the IR value, signal of a widespread damped dynamics resulting from the large anharmonicity and the abundance of soft optical modes.

Since the electrical conductivity of InSe is low (10−100 S/m), the total thermal conductivity is contributed almost entirely by the lattice portion. As shown in Fig. 4d, the thermal conductivity presents significant anisotropy in the crystal, being ~0.4 to ~9.7 W m$^{-1}$ K$^{-1}$ in the ab-plane and ~0.3 to ~2.1 W m$^{-1}$ K$^{-1}$ along the out-of-plane direction from 5 to 300 K. It can be attributed to the much weaker interlayer bonding than the intralayer ones. Above 150 K, both the in-plane and the out-of-plane thermal conductivity follow the $T^{-1}$ law, suggesting a dominant Umklapp process at high temperatures. Typically for a crystal, boundary scattering is the dominant scattering mechanism at low temperature and $\kappa_L$ roughly follows the $T^3$ scaling. Meanwhile, a $T^2$ dependence is usually observed for glasses[59]. For the inset of Fig. 4d, the $\kappa_L$-$T$ relation significantly deviates from the $T^3$ law below 50 K, instead being $T^{1.6}$ and $T^{1.4}$ for the in-plane and out-of-plane cases, respectively. This deviation suggests more complicated scattering processes of InSe. Importantly, similar deviations have been observed in several thermoelectric materials and are attributed to highly anharmonic rattling dynamics[26]. This resonates with the idea that strongly damped dynamical modes can induce glass-like properties without necessarily following from 3D structural disorder.

Finally, let us notice how the thermal conductivity in InSe does not present a sharp peak nor a Debye $T^3$ scaling at low temperature. Moreover, the thermal conductivity is quite flat against temperature, although this flattening appears at scale larger than the Boson-peak-like feature. This is different from typical glasses where the two phenomena are correlated. This further implies that thermal transport in crystalline InSe follows neither typically crystalline materials nor ideal glasses. In this sense, we propose the intepretaion that InSe should fall into the category of a crystalline system with a deviation from the Debye behaviour, somewhat a partially glassy behavior. This glassy-like physics seems much more pronounced in the out-of-plane dynamics, where thermal transport is suppressed by an order of magnitude with respect to the in-plane one. We hereto tentatively conclude that the deviation from the Debye behaviour of InSe probably originates from the unidirectional breaking of long-range order by the interlayer slip, which causes the emergence of the soft optical mode, the damped acoustic mode, the Boson-peak-like feature, and the suppressed thermal transport.

In summary, the correlation between plastic interlayer slip, structural disorder along $c$-axis, lattice anharmonicity, and thermal transport in InSe vdW crystals is comprehensively analyzed by neutron scattering and theoretical calculation. The existence, direction and magnitude of the interlayer plastic slip are validated and quantitatively described by the shift of Bragg peak positions in the (K-KL) plane. The INS measurements reveal drastic softening, broadening, and even damping of the low-energy phonons and intense acoustic-optical phonon interactions. Such an instability of this mode probably constitutes the dynamical origin for the plastic interlayer slip. Furthermore, the plastic slip induces dense stacking disorder, which in turn amplifies the lattice disorder and phonon anharmonicity, finally inducing a strongly damped ZA mode. The damping feature is supported by the discrepancy between the AIMD calculations and experimental phonon dispersions, the absence of a well-defined peak for the ZA mode, the short lifetime falling below the Ioffe-Regel limit, and the agreement between the measured and calculated heat capacities when excluding the ZA mode. As a result, the out-of-plane thermal conductivity is as low as ~0.8 W m$^{-1}$ K$^{-1}$ and significantly deviates from the Debye $T^3$ law. This work provides physical insights into the mechano-thermo relation for 2D vdW materials, which will promote the design and development of 2D materials and crystals for flexible, deformable, and shape-conformable applications in a variety of industries including energy, information, and health.

## Methods
### Crystal growth
The InSe crystal was grown by the Bridgeman method[5]. The polycrystals were firstly synthesized from 5 N purity In and Se elements with a non-stoichiometric mole ratio of In:Se = 0.52:0.48. The raw materials were sealed in a quartz ampoule under ~10$^{-3}$ Pa and then placed into a rocking furnace with a temperature of 800 °C. The raw materials were melted and soaked followed by rocking for 30 minutes. InSe polycrystals were obtained after the natural cooling of the rocking furnace. For single crystal growth, the polycrystals were re-sealed in a vacuumed quartz ampoule with the pressure less than 10$^{-3}$ Pa. The crystal growth was carried out in a custom-designed 3-zone vertical Bridgman furnace. The high-temperature zone (~660 °C) was used for material melting, the temperature gradient zone (10-15 °C/cm) was used for single crystal growth and the low-temperature zone (300-550 °C) was used for crystal annealing. Crystal growth was executed with a speed of 0.5 mm/hour for 11 days, and finally annealed for 8 hours to complete. The crystal used in neutron scattering experiments has a dimension of about 30 mm × 20 mm × 5 mm.

### Characterization
The single-crystal X-ray diffraction (XRD) was carried out using a Xtal-LAB Synergy, Dualflex, Hypix single-crystal X-ray diffractometer at room temperature. The CRI Data were measured using $\omega$ scans using Mo K$_\alpha$ radiation. The total number of runs and images was based on strategy calculation from the program CrysAlisPro 1.171.43.104a (Rigaku OD, 2023). Data reduction was performed with correction for Lorentz polarization. Numerical absorption correction is based on Gaussian integration over a multifaceted crystal model. Empirical absorption correction is conducted using spherical harmonics, implemented in SCALE3 ABSPACK scaling algorithm. The selected area electron diffraction (SAED) was conducted on a TEM (FEI Tecnai G2 F20). The accelerating voltage is 200 kV. The room-temperature Raman spectra were collected by inVia™ Raman microscope (Renishaw®, U.K.) using the excitation wavelengths of 532 nm. The temperature-dependent Raman spectra was performed in a homebuilt closed-cycle optical cryostat down to 1.6 K. A He/Ne laser centered at 633.1 nm was employed as the excitation source. A combination of one reflective Bragg grating and two Bragg notch filters allowed measurements down to −5 cm$^{-1}$. The laser power was kept below 50 μW to prevent significant laser heating of the samples. A Quantum Design Physical Property Measurement System (Quantum Design PPMS®, U.S.) was applied to measure the specific heat and thermal conductivity from 2 K to 300 K and 5 K to 300 K, respectively by a Helium-4 probe.

### Elastic neutron diffraction measurements
Elastic neutron single-crystal diffraction measurements were performed with the Wide-Angle Neutron Diffractometer, WAND[2], spectrometer at Oak Ridge National Laboratory. The neutron wavelength is 0.95 Å (Ge 115). A InSe single crystal was aligned in [HHL] plane and mounted on an Al-plate. A series of spectra were measured at $T$ = 10, 100, 200, and 300 K. The data were visualized and analyzed with the Data Analysis and Visualization Environment (DAVE)[60].

## Inelastic neutron scattering measurements

Inelastic neutron scattering (INS) measurements were performed on InSe single crystals with the MERLIN spectrometer at ISIS, UK[61]. The incident energy, $E_i$, was set as 22 meV with the energy resolution, $\Delta E/E_i \approx 5\%$ FWHM at the elastic line. Linewidths from INS are corrected by *tobyfit* in Horace taking into account the broadening of the data arising from the resolution of the instrument[62]. A single crystal of InSe with a mass about 4 g was mounted in the top-loading closed-cycle refrigerator with the crystal [1-10] direction vertical, providing the (HHL) scattering plane. Multiple data were acquired for different orientations of the crystal. The measurements were carried out at 50 K and 220 K. The data for different rotations were then combined in software to produce a four-dimensional sampling of the scattering function $S(q, E)$ of the crystal. Small departures from the exact horizontal (HHL) orientation were corrected in the software. The data were analyzed with HORACE software package[62]. Figure 2c-d are intensity plots considering the calculated dynamical structure factor from AIMD and instrument resolution.

## Calculation methods

First-principles simulations were performed in the framework of density function theory (DFT) as implemented in the Vienna ab initio simulation package (VASP)[63,64] with the projector-augmented wave (PAW) method[65]. The r²SCAN meta-GGA functional[66] is employed in this work due to its improved numerical performance. The valence configurations are taken as In: $5s^25p^1$, Se: $4s^24p^4$. We optimized the unit cell by setting a plane-wave energy cutoff of 520 eV, an energy convergence criterion of $5 \times 10^{-8}$ eV and force convergence criterion of $1 \times 10^{-5}$ eV/Å. The **k**-point mesh was set to $12 \times 12 \times 3$. The ab initio molecular dynamics (AIMD) simulations were initially conducted using the isothermal-isobaric ensemble (NPT), employing a Langevin thermostat for a duration of 6 ps with a time step of 2 fs. Subsequently, the simulations were performed using the canonical ensemble (NVT), employing a Nose-Hoover thermostat for 20 ps with a time step of 2 fs across a range of temperatures: 50 K, 200 K, and 400 K. The phonon dispersions were calculated in a $5 \times 5 \times 2$ supercell (400 atoms). The second interatomic force constants (IFCs) were fitted with random displacements generated by the Monto-Carlo rattle procedure in the hiPhive package[67], with a cutoff radius of 10.2 and 6 Å for the second- and third-order force constants, respectively. Based on the force constants, the phonon dispersions and phonon lifetimes used in this work are generated by Phonopy[68,69] and ShengBTE package[70], respectively, based on the second- and third-order interatomic force constants obtained from the AIMD results. Furthermore, the dynamical susceptibility $\chi''(q, E)$ were simulated with the software Euphonic[71], also based on the AIMD results.

## Data availability

The data that support the findings of this study are available from the corresponding author upon request. Raw data from the neutron scattering experiment are available here: https://doi.org/10.5286/ISIS.E.RB1920240.

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

## Acknowledgements

This work is supported by National Key Research and Development Program of China (Grant No. 2022YFA1402702), the National Science Foundation of China (Nos. T2122013, U2032213, 52272006), and the interdisciplinary program Wuhan National High Magnetic Field Center (Grant No. WHMFC 202122). M.B. acknowledges the support of the Shanghai Municipal Science and Technology Major Project (Grant No.2019SHZDZX01) and the sponsorship from the Yangyang Development Fund. J. Yang acknowledges the support of the Key Research

Project of Zhejiang Lab (No. 2021PE0AC02). M.J. acknowledges the support of the Space Application System of China Manned Space Program. M.F.S thanks the support from Guangdong Provincial Key Laboratory of Extreme Conditions (Grant No. 2023B1212010002). M.B. acknowledges the support of the Shanghai Municipal Science and Technology Major Project (Grant No.2019SHZDZX01) and the sponsorship from the Yangyang Development Fund. The single crystal XRD work at MSU was supported by U.S.DOE-BES under Contract DE-SC0023648. Experiments at the ISIS Neutron and Muon Source were supported by a beamtime allocation RB1920240 from the Science and Technology Facilities Council. Data is available here: https://doi.org/10.5286/ISIS.E.RB1920240.

## Author contributions

T.R.W. and J.M. designed the project. Y.F. Liu, Y.P.M., M.J., and T.R.W. prepared the samples and performed the electron diffraction and Raman measurements. H.Z.W. and W.W.X. performed the single crystal X-ray diffraction measurement. S.W.J., P.F.J., and J.M. performed the temperature-dependent Raman measurements. J.T.W. and Y.Liu carried out the transport measurements. Y.Lin and J.Y. performed the calculations. J.T.W., G.T.L., M.F.S., C.P.Z., F.F.Z., Y.W., R.E., H.W., G.C.D., S.X.C., and J.M. performed the INS experiment and neutron diffraction experiments. J.T.W., M.F.S., M.B., T.R.W., J.Y., X.S., and J.M. analyzed the data and wrote the manuscript. All authors reviewed and edited the manuscript.

## Competing interests

The authors declare no competing interests.

## Additional information

[1]Key Laboratory of Artificial Structures and Quantum Control, School of Physics and Astronomy, Shanghai Jiao Tong University, Shanghai 200240, China. [2]Materials Genome Institute, Shanghai University, 99 Shangda Road, 200444 Shanghai, China. [3]State Key Laboratory of Metal Matrix Composites, School of Materials Science and Engineering, Shanghai Jiao Tong University, Shanghai 200240, China. [4]State Key Laboratory of Functional Materials for Informatics, Shanghai Institute of Microsystem and Information Technology, Chinese Academy of Sciences, 200050 Shanghai, China. [5]Neutron Scattering Division, Oak Ridge National Laboratory, Oak Ridge, TN 37831, USA. [6]ISIS Pulsed Neutron and Muon Source, STFC Rutherford Appleton Laboratory, Harwell Campus, Didcot OX11 0QX, United Kingdom. [7]Australian Centre for Neutron Scattering, Australian Nuclear Science and Technology Organisation, Lucas Heights, NSW, Australia. [8]Wilczek Quantum Center and School of Physics and Astronomy, Shanghai Jiao Tong University, Shanghai 200240, China. [9]College of Materials, Shanghai Dianji University, Shanghai 201306, China. [10]Department of Chemistry, Michigan State University, East Lansing, MI 48824, USA. [11]State Key Laboratory of High Performance Ceramics and Superfine Microstructure, Shanghai Institute of Ceramics, Chinese Academy of Sciences, Shanghai 200050, China. [12]Collaborative Innovation Center of Advanced Microstructures, 210093 Nanjing, Jiangsu, China. [13]These authors contributed equally: Jiangtao Wu, Yifei Lin, Mingfang Shu. ✉e-mail: tianran_wei@sjtu.edu.cn; jiongy@t.shu.edu.cn; jma3@sjtu.edu.cn

