## [Peer Review File · Nature Communications]

Uncovering the phonon spectra and lattice dynamics of plastically deformable InSe van der Waals crystalsREVIEWER COMMENTS

Reviewer #1 (Remarks to the Author):

J. Wu et al present a detailed study of the atomic structure and dynamics in InSe, a layered van-der-Waals crystal. On the experimental front, they present diffraction, spectroscopic and thermophysical data using a variety of techniques, including electron and neutron diffraction, Raman scattering, and inelastic neutron scattering. These data are complemented via the calculation of phonon band structures, paving the way for a microscopic interpretation of the experiments. On the basis of these results, the authors identify a manifold of low-energy modes which are linked to departures of low-temperature thermophysical properties from expectation, particularly the heat capacity. The out-of-plane transverse phonon mode is identified as the one undergoing the most significant changes, and these effects are given a physical interpretation in terms of nesting effects (also seen in other systems) as well as what the authors call a ‘frequency resonance.’ These new findings are placed in a wider context as being responsible for the plasticity exhibited by this van-der-Waals crystal in its mechanical (macroscopic) properties.

Overall, the work presented is quite comprehensive and certainly merits publication – possibly in a journal with the impact and scope of Nature Communications, given its wider relevance within the remit of our current (and quite limited) understanding of the properties of soft layered materials. The use of single-crystal specimens, for example, has enabled a rather elegant determination of the stacking sequence and the short-range order along the c-axis, both linked to the interlayer slip path. The authors also make an attempt to relate the inelastic neutron scattering results to calculations of phonon band structures, including changes relative to a so-called ‘perfect crystal’ and their impact on thermophysical properties. As presented, I find this latter link far more tenuous and unclear than the other results. On the basis of the above, I would encourage the authors to embark on a significant revision of the present manuscript to address these shortcomings, as reflected in the following and more detailed remarks:

- Title: I am not sure that, as written, the title conveys the primary message(s) that the authors are trying to convey, particularly to a less-specialized scientific audience. Likewise, I am not convinced that the word ‘mediated’ is the most appropriate one to describe a physical picture that is consistent with the experimental results.
- Lines 152-154, “Furthermore, the slipped structure corresponding to the slip vector $(1/3, 2/3, 0)$ shows a comparatively low energy compared to the unslipped structure.” It is not clear how these differences in energy have been inferred from the data, as well as how large these are. Please explain.
- Figure 1: this figure and the accompanying discussion in the main text do not make it clear at what temperature these neutron data were taken – presumably either at 50 K or 200 K. Are there any differences between these two datasets, particularly in terms of the diffuse scattering?

- Lines 172-173, the manuscript states that “...showing excellent agreement for both the phonon energies and INS intensities. However, the observed phonon dispersions deviate from the AIMD calculations for the low energy phonons. Particularly, the out-of-plane transverse acoustic branch (denoted as ZA, and similar to that in 2D materials) with the lowest energy seems to be missing in the experimental data.” Visual inspection of Fig. 2 is certainly not sufficient to verify to what extent these statements hold, as well as what the deviations are at the lower energies, particularly in terms of the absence of the ZA mode. I would suggest presenting additional figures and/or data where these effects are more evident.
- Figure 2: the term ‘perfect crystal’ is used when referring to the calculations. Please clarify what is meant by this term and how this model might differ from the real material. Likewise, are there any reasons not to use as starting point for the calculations the information inferred from the analysis of elastic, diffuse and Raman scattering data presented earlier?
- Figure 2 and accompanying Supplementary Material. This figure reports data at 200 K, yet there is another dataset at 50 K. One would expect comparison with calculated data to be more direct using the dataset at the lower temperature, particularly in view of the meV energy scales probed in this work – ideally, this would require an inelastic neutron scattering measurement at very low temperatures, which does not seem to have been performed. Please explain.
- Line 198: the concept of ‘nesting’ is introduced, yet I am not sure that the brief explanation given about its physical meaning and significance will be of much use beyond the specialist. I would encourage the authors to provide a more transparent explanation, for the benefit of the wider audience.
- Figures 3c and 3d: some linewidths go to zero at the higher q ’s, and some others seem to be at the limit of the experimental resolution explained in the Supplementary Information. Also, the Supplementary Information indicates that these have been obtained from fits using Gaussian line shapes. This is a surprising choice: physically, one would expect that spectral features would conform to a Lorentzian for homogeneous broadenings, or to a Voigt if resolution effects and/or additional sources of inhomogeneous broadening are important. Likewise, Fig. 4c shows that experimentally derived phonon linewidths remain quite constant over a wide temperature range and they are well below computational predictions up to mode energies of around 10 meV. This behavior is quite counterintuitive. Please explain.
- Figure 4a: the experimental work has measured constant-pressure heat capacities C_p , while the expressions used to obtain the heat capacity from the calculations appear to be the isochoric counterpart C_v . These two quantities are certainly not the same. Please elaborate on this point, including a discussion of they can be related to each other quantitatively for this particular material.
- Figure 4b and accompanying discussion in the main text. The authors state that “These results suggest that the presence of damped low energy optical phonon modes is responsible for the

Boson peak anomaly ...” I can see how the presence of a dense manifold of low-energy phonon modes can lead to the C/T^3 anomaly reported in Fig. 4b. Yet these modes can still be perfectly harmonic and they can still give rise to these features in the specific-heat data – i.e., the presence of mode energies in the appropriate range is a sufficient condition to see deviations from Debye’s law. I do not understand why damping (i.e., finite lifetimes) needs to be invoked in order to account for it.

- Figure 4d and Supplementary Material: I could not find a detailed discussion of the thermal conductivity data reported in this figure. Please include more information.

- The word ‘collapse’ is only used in the abstract when referring to the out-of-plane transverse phonon mode. In the subsequent discussion (line 174), the manuscript states that this mode ‘seems to be missing in the experimental data.’ Removal of this mode from the calculation of the heat capacity seems to help reconciling experimental data and computational predictions, which is encouraging – Fig. 4b and lines 241-243. I found the physical arguments put forth to explain these observations quite confusing to follow – i.e., where does this mode (presumably propagating?) go when transitioning from the ‘perfect crystal’ to the real material? Physically speaking, ‘disappearance’ does not seem to be the right term to use, given the general (normalization) properties of the underlying density of states.

- In line with the above, the abstract also refers to a ‘frequency resonance’ effect which does not seem to be explained in any depth in the subsequent discussion.

- Throughout the manuscript, there is a limited description of the possible role of other defects on experimental observables, particularly those of a thermophysical nature at low temperatures. This point should be addressed in more depth, particularly when singling out interlayer slip as the only mechanism at play in this material.

- Description of the calculations, in the main text and in the Supplementary Material: I found the description of these quite confusing and fragmented all throughout – it needs to be improved considerably. The authors certainly recognize the value of carrying out both harmonic lattice-dynamics calculations as well as AIMD, yet they do not seem to capitalize from their complementarity. To note:

- o Supplementary Info, Fig S3: have these been obtained from harmonic calculations? Under the quasi-harmonic approximation or by relaxing unit cell to its internal-energy minimum? Clarify.

- o Supplementary Info, Fig S4: how do these AIMD calculations compare to those of Fig. S3? A comparison at the level of the phonon DOS (possibly including various projections) should already be telling you something about anharmonic effects, for example.

- Throughout the text: terms and expressions such as ‘giant anharmonicity’ or ‘collapse of the out-of-plane transverse phonon mode’ would need to be put on more quantitative grounds for them to be meaningful – e.g., ‘giant anharmonicity’ compared to what?

Reviewer #2 (Remarks to the Author):

The authors performed diffraction, diffuse scattering, and inelastic measurements on vdW crystal samples of InSe, which is a material with exceptional plasticity, using electron diffraction and neutron scattering. The measurements of phonon dispersion, heat capacity, phonon lifetime, and thermal transport were compared with ab initio molecular dynamics simulations. They reveal some interesting physics, in particular, the relation between layer slip and the phonon anharmonicity. The manuscript will be suitable for publication if the authors address the following issues.

1. The rods along the 001 direction in the diffuse scattering suggests the loss of long-range order. On the other hand, the explanation on the relative slip in the reciprocal space is confusing. Such “slip” indicates correlation between the slips in real space. Does this indicate that the overall lattice takes a different lattice structure from 2H?
2. The missing ZA mode in Fig. 2 needs better labeling. It is difficult to be identified in the existing plots. The difference between the experiment and simulation is not very clear.
3. It would be nice to compare the optical phonon eigenvectors with the slip directions.
4. Why is the localization of the optical modes not observed by the inelastic neutron scattering? They shall show as dispersionless signal through the Brillouin zones.
5. Is the instrument resolution subtracted from the linewidth?
6. One of the most important observations of the work is the deviation of the Cp from the Debye model and the Boson peak (they are the same feature and shall not be discussed differently). The other important observation is the loss of the ZA phonon model. The manuscript shall focus more on this.

Reviewer #3 (Remarks to the Author):

The present work is reporting the lattice dynamics properties of beta-InSe from inelastic neutron scattering (INS), heat capacity and thermal conductivity experiments on single-crystalline samples and ab initio calculations and its connection with interlayer slip characterizing with electron and neutron diffraction experiments.

Notably, the INS experiments show very large phonon dispersion for the two low energy optical phonon branches between 2 and 6 meV and between 5 and 8 meV with significant anharmonicity close to Brillouin zone center. These results show unusual vibrational dynamics linked with interlayer slip, which is quite unique and of high interest for the field of the thermoelectric materials and of 2D van der Waals layered chalcogenides.

These interesting results are analyzed with the help of ab initio calculations but they must be very much improved as there are many uncertainties with them as will be discussed and there are not able to explain the results of thermal conductivity. Also, the discussion on the heat capacity seems unsatisfactory.

Therefore, if the INS experiments are of high interest, potentially publishable in Nature Communications, the present work is still far from the high standard of Nature Communication. However, I am convinced that the present work could perhaps be published in Nature

Communication after a major revision following the comments detailed below.

1) Structure characterization

The authors did not give the crystal structure obtained from their diffraction data. They must do it. Also, the authors did not inform the reader of the existence of the epsilon phase with the same lattice parameters than the beta phase but with slightly different crystal structure, they must do it. I don't think that the use of Raman scattering for distinguishing between the beta and epsilon phases is enough. I think that additional calculations are needed for confirming that. See the calculations for more details.

2) Inelastic neutron scattering experiments

The authors have analyzed their data using gaussian functions. However, giving the broad linewidth of the observed lines, it seems that the damped oscillator model should be better adapted and therefore to use lorentzian functions instead of gaussian functions. I think that the HEB is also interesting and the energy dispersion must be shown as for the LEB in Fig. 3(a) and (b), at least in the supplementary informations. Concerning the linewidth of such HEB, as it seems to correspond to two different modes, it is maybe not needed to show them, but this could be still interesting to show them. The discussion in lines 188-191 must be revised because it is unaccurate. Indeed, the authors must take more care when comparing with SnSe and PnTe/SnTe. Indeed, in SnSe the phonon dispersion is similar in the Gamma-Y direction but not in the other directions and it experienced phase transition at 800 K (see ref. 14).

Despite that, the energy of low energy TO mode decreases with increasing T, which is not the behavior of a soft mode.

On the other hand, the TO mode of SnTe and PbTe have a soft mode behaviour with their energies increasing with increasing T (refs. 15). This is a different behavior than in InSe, where it has more normal behavior with energies decreasing with increasing T.

Therefore, the discussion in lines 188-191 must be modified.

In the figure 3 (c), the authors have added the q variation of the linewidth of SnTe, PbTe and SnSe from the literature. There are some problems with these data.

For PbTe, in ref. 16 (also in ref. 19), the data correspond to acoustical phonons and not optical mode and therefore this is nonsense to compare them with the results obtained for InSe.

For SnSe, the data in ref. 17 correspond to Na doped SnSe and not pure SnSe and it is not clear how the data plotted in fig. 3(c) have been obtained.

For SnTe, the data in ref. 18 looks to have broader Q range.

Note that in the Methods part, it is uncorrectly written that the INS experiments were performed at 220 K instead of 200 K.

3) Heat capacity and thermal conductivity

I don't agree with the use of Boson peak for the Debye plot of the heat capacity because it means that it should be linked with disorder and/or glassy behavior.

In fact, the Debye model can be applied strictly on the case with only one atom where there is only acoustical phonon. In more complex case, there is always deviation from the Debye behavior because of the presence of optical modes. When there is low energy optical modes, it is well documented that there is always a peak in the Debye plot of the heat capacity both in experiments and in calculations. Therefore, the authors must remove all the paragraph from the lines 256 and 279.

There is also generally some deviations between the calculations and the experiments. It is not

surprising that there is some deviations in their case, notably because the harmonic calculation did not take into account to the large anharmonicity of the LEB and HEB and also maybe because of the disappearance of the ZA branche because of the large disorder in the c direction, as proposed by the authors.

Because of the large dispersion of the LEB and HEB, the Einstein model is not valid anymore. More sophisticated model including both the large dispersion of the HEB and LEB must be used.

However, I think that it is just simpler to remove the part on the Debye-Einstein model fitting the heat capacity data.

I don't agree with the authors that there is flattening in their thermal conductivity data. Indeed, the thermal conductivity must be plotted in a linear scale, otherwise it is more difficult to see the decrease of the thermal conductivity with increasing temperature with semi-log scale used in Fig. 4 (d).

For the out-of plane thermal conductivity, one can see that it decreases from about 2 W/m.K at about 50 K down to about 0.8 W/m.K whereas in the case of the in-plan thermal conductivity, which is rather large, it decreases from about 10 W/m.K at about 50 K to 6 W/m.K at 300 K. Therefore is maxima in the thermal conductivity, even in the case of out of plane thermal conductivity. Therefore this give some doubts about the glassy behaviour claimed by the authors. Do the authors check if the thermal conductivity is following $1/T$ dependence ?

Also, I think that in the inset of Fig. 4 (d) in log-log scale, they must show the power laws followed by the thermal conductivity.

Concerning the discussion on the thermal conductivity, I think that it must be more discussed in light of the DFT calculations. See the comments on calculations below.

4) Calculations

I don't think that the calculations are sufficiently converged. Indeed, for the structural relaxation of the primitive cell, the authors used a $4 \times 4 \times 3$ k-point mesh despite the lattice parameter a is about 4 angstroms and the lattice parameter c is about 17 angstroms. Clearly, the density of k points is too small in the a and b directions.

I would expect to use $12 \times 12 \times 3$ k-point mesh. Also, the use of $1 \times 1 \times 1$ k-point mesh for the $4 \times 4 \times 1$ supercell is certainly insufficient as well. Instead, we would expect to use $3 \times 3 \times 3$ k-point mesh. Therefore, the authors must redo their calculations with finer k-point mesh.

They must also indicate if they use Gamma centered or Monkhorst pack grids.

They must give the criterion used for the force convergence.

They must compare their calculated crystal structure with the experimental one obtained from their diffraction data

and with the literature (in the supplementary information). Concerning the phonon dispersion curves, they must show them not only along Gamma-K and Gamma-M but also in the Gamma-A, Gamma-L and Gamma-H directions. This will give very useful informations. They must also give the list of the phonon modes at Gamma point with their symmetries and energies, as Bejani et al (Phys. Rev. Mater. 6, 115201 (2022)) and compare them with their experimental results as well as the literature results (Raman and infrared spectroscopy) in the supplementary information. They must also give the symmetries and the energies of the three optical modes in figure 2(f).

As Bejani et al have performed some previous lattice dynamics calculations on beta-InSe with DFT, they must be cited and their results must be compared with the present results.

Giving the importance of the LO modes for distinguishing with Raman scattering experiments

between the beta phase and the gamma and epsilon phases, and more particularly the last one, it is essential to also calculate the LO-TO splitting. Concerning the distinction between the beta and the epsilon phases, the authors must indicate more clearly that there is a claim that beta and epsilon phases can be distinguished with the absence or presence of the Raman line at about 199 cm^{-1} which should a LO mode of the epsilon phase because the LO modes are both Raman and IR active in the epsilon phase but there was no theoretical support of this claim. Therefore, harmonic lattice dynamics calculations with DFT including LO-TO splitting must be performed for confirming this claim. This is important because every thing distinguishing the beta and epsilon phases is based on that claim that have never really checked with calculations. We must be sure that the present sample has beta structure and not epsilon structure.

The phonon lifetime must be as well calculated with finer k-point mesh. It is clear for me that the very large anisotropy of the experimental thermal conductivity must better analyzed using notably DFT calculations. For a better analysis and discussion of the results of thermal conductivity, the authors must calculate the thermal conductivity from all the previous calculations they have done and also plot the group velocities as function of the energies and/or modes. Indeed, I don't agree with their interpretation of the thermal conductivity.

Clearly, the LEB and HEB have quite large dispersion and we can expect that their group velocities must be high enough for having a large contribution to the thermal conductivity. For better seeing that, a calculation of the cumulated thermal conductivity vs energy is needed for both a and c directions and probably also from the different phonon branches.

I am convinced that one of the main reasons of the anisotropy of the thermal conductivity (but not the only one, certainly the disorder play also one important role) is the largest contribution from the optical phonons below 9 meV.

Concerning the AIMD calculations, the phonon dispersion curves are shown in Fig. S4 (a) only below 16 meV. Please show the data for the full energy scale and compared the 50 K data with the data from harmonic calculations. Indeed, at 50 K the phonon dispersion from AIMD look very different from the harmonic phonon dispersion, especially the branches between 10 and 16 meV, especially when approaching the Brillouin zone center for which the energy is much smaller in the AIMD calculations than in the harmonic calculations. Even for the lower HEB and LEB, the energies close to Gamma point in the AIMD are significantly larger than in the harmonic calculations.

Have you some explanation for these strong disagreements ? I suspect again that this is linked with the insufficient convergence conditions. Strangely, the calculated Raman spectra in Fig. S4 (b) have low energy mode at 2-2.5 meV but this does not correspond to the energies of the lowest energy mode in the phonon dispersion curves in the Fig. S4 (b). It is also not clear to which phonon mode in the phonon dispersion curves is corresponding the Raman mode at 5 meV. Could you explain this disagreement ?

Here, like for the harmonic modes, it is necessary to well identify the different vibrational modes and their symmetries. Also, it is necessary to give the method used for calculating the Raman spectra.

Responses to Reviewers' comments:

Reviewer #1 (Remarks to the Author):

J. Wu et al present a detailed study of the atomic structure and dynamics in InSe, a layered van-der-Waals crystal. On the experimental front, they present diffraction, spectroscopic and thermophysical data using a variety of techniques, including electron and neutron diffraction, Raman scattering, and inelastic neutron scattering. These data are complemented via the calculation of phonon band structures, paving the way for a microscopic interpretation of the experiments. On the basis of these results, the authors identify a manifold of low-energy modes which are linked to departures of low-temperature thermophysical properties from expectation, particularly the heat capacity. The out-of-plane transverse phonon mode is identified as the one undergoing the most significant changes, and these effects are given a physical interpretation in terms of nesting effects (also seen in other systems) as well as what the authors call a frequency resonance; These new findings are placed in a wider context as being responsible for the plasticity exhibited by this van-der-Waals crystal in its mechanical (macroscopic) properties.

Overall, the work presented is quite comprehensive and certainly merits publication possibly in a journal with the impact and scope of Nature Communications, given its wider relevance within the remit of our current (and quite limited) understanding of the properties of soft layered materials. The use of single-crystal specimens, for example, has enabled a rather elegant determination of the stacking sequence and the short-range order along the c-axis, both linked to the interlayer slip path. The authors also make an attempt to relate the inelastic neutron scattering results to calculations of phonon band structures, including changes relative to a so-called 'perfect crystal' and their impact on thermophysical properties. As presented, I find this latter link far more tenuous and unclear than the other results. On the basis of the above, I would encourage the authors to embark on a significant revision of the present manuscript to address these shortcomings, as reflected in the following and more detailed remarks:

Response: We are grateful to the Reviewer for recognizing the comprehensive character and the merits of our work and for assessing that our results are of wider relevance within the remit of our current (and quite limited) understanding of the properties of soft layered materials. We also thank the Reviewer for the constructive comments which resulted extremely helpful in improving this article. Below we provide

our detailed responses to each of the comments.

1. Title: I am not sure that, as written, the title conveys the primary message(s) that the authors are trying to convey, particularly to a less-specialized scientific audience. Likewise, I am not convinced that the word ‘mediated’ is the most appropriate one to describe a physical picture that is consistent with the experimental results.

Response: We appreciate the Reviewer for this suggestion and we agree that the word ‘mediated’ might not be appropriate to describe the physics. We revised the title as “Overdamped Phonons: Uncovering the Consequences of Interlayer Slip in 2D van der Waals InSe Crystals”.

2. Lines 152-154, “Furthermore, the slipped structure corresponding to the slip vector $(1/3, 2/3, 0)$ shows a comparatively low energy compared to the unslipped structure.” It is not clear how these differences in energy have been inferred from the data, as well as how large these are. Please explain.

Response: We thank the reviewer for this comment. The energy difference between the initial and slipped structure ($E_{\text{slip}} - E_{\text{initial}}$) is calculated to be nearly zero by previous studies (2D Mater., 2021, 8, 45028; Nat. Mater., 2024, 23, 196-204). Accordingly, we have revised the sentence as:

“Furthermore, the slipped structure corresponding to the slip vector $(1/3, 2/3, 0)$ shows a comparatively low energy compared to the unslipped structure; the energy difference is nearly zero according to previous calculations.^{3,16}”

3. Figure 1: this figure and the accompanying discussion in the main text do not make it clear at what temperature these neutron data were taken – presumably either at 50 K or 200 K. Are there any differences between these two datasets, particularly in terms of the diffuse scattering?

Response: We thank the referee for the comment. In order to keep the neutron diffraction and inelastic neutron scattering data at the same temperature, the neutron data in Fig. 1 were obtained at 200 K. Except for the thermal effect on the intensity, there is no big difference on the diffuse scattering between 50 K and 200 K (Figure R1). As the statistics of phonon data at 50K are worse than the dataset at 200K for Fig. 2, we decided to use the dataset at 200K to keep the dataset at same temperature.

Figure R1. The comparison of the diffuse scattering of 2H(β)-InSe at 200K (a) and 50K (b).

4. Lines 172-173, the manuscript states that “...showing excellent agreement for both the phonon energies and INS intensities. However, the observed phonon dispersions deviate from the AIMD calculations for the low energy phonons. Particularly, the out-of-plane transverse acoustic branch (denoted as ZA, and similar to that in 2D materials) with the lowest energy seems to be missing in the experimental data.” Visual inspection of Fig. 2 is certainly not sufficient to verify to what extent these statements hold, as well as what the deviations are at the lower energies, particularly in terms of the absence of the ZA mode. I would suggest presenting additional figures and/or data where these effects are more evident.

Response: We thank the referee for the comment. The evidences of the absence of the ZA mode are: 1) In Fig.3, the constant-q cut of LEB phonon band shows only one signal instead of two acoustic phonons as TA and ZA; 2) in Fig.4(b), the fitted C_p without ZA contribution agrees with C_p measurement. In order to clarify the difference between the experiment and simulation, we plotted the simulated ZA as a dashed-line(Figure R2); 3) in Fig.4(c) the lifetime τ_{exp} falls below the IR limit for the low-energy phonons but exceeds this limit for high-energy phonons. The evidences might only confirm the overdamped nature of ZA, not be enough to conclude the absence of the ZA mode, hence, we revised the manuscript as “...showing excellent agreement for both the phonon energies and INS intensities. However, the observed phonon dispersions deviate from the AIMD calculations for the low energy phonons. Particularly, the out-of-plane transverse acoustic branch (denoted as ZA with dash-lines, and similar to that in 2D materials) with the lowest energy seems to be strongly overdamped in the experimental data.”

Figure R2. The phonon dispersions observed in the dynamical susceptibility $\chi''(Q, E)$ from (a),(b) inelastic neutron scattering (INS) measurements at 200 K. The dash-lines are the simulated ZA phonon dispersions by the *ab initio* molecular dynamics (AIMD) simulation at 200 K.

5. Figure 2: the term ‘perfect crystal’ is used when referring to the calculations. Please clarify what is meant by this term and how this model might differ from the real material. Likewise, are there any reasons not to use as starting point for the calculations the information inferred from the analysis of elastic, diffuse and Raman scattering data presented earlier?

Response: We thank the referee for this comment. The “perfect crystal” corresponds to the ideal case considering that the crystal is stacked exactly following the pre-assumed structure (2H phase in this work) without any defects such as dislocations, stacking faults, vacancies. Of course, such a “perfect structure” is different from the real crystals. Here in this work, we find that the interlayer slip is quite easy to happen, which interrupts the ideal structure to some extent and introduces some defects like stacking faults. The difference between the experimental data on the “real crystal” and the simulation data of the “perfect crystal” could tell the effect of the interlayer slip. In the simulation, the starting point of the lattice constants, space group, atomic positions were obtained from the analysis of elastic, diffuse and Raman scattering data.

As the reviewer pointed out, it would be better if we construct the structure for the “real crystal” based on the experimental data. However, this is almost impossible since we cannot fully describe the type and distribution of these abundant defects induced by the interlayer slip. In order to avoid the misunderstanding, the term “perfect crystal” has been removed.

6. Figure 2 and accompanying Supplementary Material. This figure reports data at 200 K, yet there is another dataset at 50 K. One would expect comparison with calculated data to be more direct using the dataset at the lower temperature, particularly in view of the meV energy scales probed in this work – ideally, this would require an inelastic neutron scattering measurement at very low temperatures, which does not seem to have been performed. Please explain.

Response: We thank the referee for the suggestion. Due to the thermal factor, the signal intensity and statistics of phonon data at 50K are not as good as the data set at 200 K(Figure R3). As there is no phase transition between 50K and 200K, we used 200K for Figure 2. In the revised version, we have included the INS data along [K-K0] at 50K in Fig.S4(c) SI.

Figure R3. The comparison of the inelastic neutron scattering of β -InSe at 50 K (a, b) and 200 K (c, d), respectively.

7. Line 198: the concept of ‘nesting’ is introduced, yet I am not sure that the brief explanation given about its physical meaning and significance will be of much use beyond the specialist. I would encourage the authors to provide a more transparent explanation, for the benefit of the wider audience.

Response: We thank the referee for this suggestion. The nesting effect usually behaves as the parallel phonons and presents as phonon-phonon interaction. We have revised

the manuscript as “Unlike augmenting the three-phonon scattering channels SnTe/PbTe mainly around zone center, the local phonons of InSe present “nesting” behavior with two paralleled phonon groups over a large q-range and enable more acoustic-optical three-phonon scattering channels which amplifies the anharmonicity.¹⁴”

8. Figure R 3c and 3d: some linewidths go to zero at the higher q's, and some others seem to be at the limit of the experimental resolution explained in the Supplementary Information. Also, the Supplementary Information indicates that these have been obtained from fits using Gaussian line shapes. This is a surprising choice: physically, one would expect that spectral features would conform to a Lorentzian for homogeneous broadenings, or to a Voigt if resolution effects and/or additional sources of inhomogeneous broadening are important. Likewise, Fig. 4c shows that experimentally derived phonon linewidths remain quite constant over a wide temperature range and they are well below computational predictions up to mode energies of around 10 meV. This behavior is quite counterintuitive. Please explain.

Response: We thank the referee for the comment. It is important to subtract the instrument resolution from the measured linewidth, but it is very challenge for a time-of-flight instrument. We have used two methods: 1) the software of Tobyfit; 2) fitted the elastic peak by Gaussian function as the instrument resolution and subtracted the resolution from the Lorentzian fitting. Although the linewidth values of those two methods are little bit different, the tendencies of q-dependent linewidths from those methods are similar. After discussed with the instrument scientists, we decided to use the values from the software of Tobyfit. Unfortunately, there was only Gaussian function in Tobyfit in the submitted draft. We have updated the code of Tobyfit to include the Lorentzian function. In the revised manuscript, the linewidths have been updated by the Lorentzian fitting.

In Fig.3(c) and Fig.3(d), the TA, ZA and TO are strongly interacted/correlated at the higher q and it is resolution limited form the software of Tobyfit, hence, it is hard to obtain the phonon linewidth.

In Fig.4(c), the energy-dependence of the lifetime increases by the fitting, which is different to the decreasing by the simulation. The constant values might be due to the lg-lg plot, which we want to present the ω^{-2} relation. In order to state the experiment data, we have added one more sentence in the caption of Fig.4 and the text, “The

lifetimes increase with phonon energies. ”

9. Figure 4a: the experimental work has measured constant-pressure heat capacities C_p , while the expressions used to obtain the heat capacity from the calculations appear to be the isochoric counterpart C_v . These two quantities are certainly not the same. Please elaborate on this point, including a discussion of they can be related to each other quantitatively for this particular material.

Response: We thank the reviewer for this suggestion. The measured C_p is contributed by: i) harmonic phonons ($C_{ph,H}$), ii) lattice dilation (or anharmonic phonons, C_D), and iii) electrons (C_{el}). Since the carrier concentration and electrical conductivity is extremely low in InSe, the electronic contribution can be safely excluded. Then,

$$C_p = C_{ph,H} + C_D$$

Since the contribution of phonons in the harmonic regime does not depend on volume, so the constant-pressure and constant-volume results are identical:

$$C_{ph,H} = C_{ph,H}|_P = C_{ph,H}|_V = C_v = C_{AIMD}$$

which is exactly the value obtained by *ab initio* molecular dynamics calculation (detailed in SI).

According to references (Phys. Rev. B, 2009, 80, 184302), lattice dilation C_D can be obtained from

$$C_D(T) = 9Bv\alpha^2T$$

With this notation

$$C_p = C_v + 9Bv\alpha^2T$$

where B is the isothermal bulk modulus (26.007 GPa [Nat. Mater., 2024, 23, 196]), v is the molar volume ($1.439 \times 10^{-4} \text{ m}^3/\text{mol}$), and α is the linear coefficient of thermal expansion (Solid State Commun., 2021, 326, 114163). The calculated data are shown here below in Figure R4, it is clear that the difference between C_p and C_v is less than 6 % from 30 K to 300 K.

The above discussion has been included in the revised manuscript. In the main text, we also mention that “Since the lattice dilation effect and the electronic contribution to heat capacity are trivial, it is safe to treat C_p as C_v for modelling, at least below 200 K.”

Figure R4 The heat capacity with varied temperatures.

10. Figure 4b and accompanying discussion in the main text. The authors state that “These results suggest that the presence of damped low energy optical phonon modes is responsible for the Boson peak anomaly ...” I can see how the presence of a dense manifold of low-energy phonon modes can lead to the ‘ C/T^3 anomaly’ reported in Fig. 4b. Yet these modes can still be perfectly harmonic and they can still give rise to these features in the specific-heat data – i.e., the presence of mode energies in the appropriate range is a sufficient condition to see deviations from Debye’s law. I do not understand why damping (i.e., finite lifetimes) needs to be invoked in order to account for it.

Response: We thank the reviewer for this comment and agree with the reviewer technically. In Fig.4(b), the C_p/T^3 was calculated from the phonon density-of-states at 200 K by AIMD through the harmonic model, it is larger than the measurement. A soft optical mode is enough to have a peak, for example Einstein mode, nevertheless, the broadening of the peak and the previously mentioned overdamped nature of the ZA suggest that the Boson-peak-like signal in this material is associated also to strongly anharmonic mode, hence we emphasized “damping”. The optical mode energies of a perfectly harmonic system are always above the energy of the acoustic modes at the Brillouin zone boundary, so the low optical phonons are also the signatures of strong interactions etc. Combining with previous discussion on the damped ZA, we subtracted the ZA contribution from the total C_p/T^3 directly, and find that the agreement between the subtracting C_p/T^3 and the experimental value is good. All calculations are from the harmonic model, but the method of subtracting ZA is the anharmonic analysis on the phonons (damped ZA). Although treating the anharmonicity by excluding the damped-phonon is simplified, we tried to give a suggestion on the explanation of the Boson-peak-like feature. In order to clarify the statement, we have revised the text,

“As shown in Fig. 4(b), the ZA contribution is tentatively subtracted from the total phonon DOS, $C_{AIMD-tot-ZA}/T^3$. Since the calculated results agree with the measurement, the treatment of the overdamped ZA mode is expected for the macroscopic thermal property.”

11. Figure 4d and Supplementary Material: I could not find a detailed discussion of the thermal conductivity data reported in this figure. Please include more information.

Response: We thank the referee for this suggestion. We have added some discussion on thermal conductivity in the revised manuscript.

“As shown in Fig. 4(d), the thermal conductivity presents significant anisotropy in the crystal, being ~ 0.4 to ~ 9.7 W/m·K in the ab-plane and ~ 0.3 to ~ 2.1 W/m·K along the out-of-plane direction from 5 to 300 K. It can be attributed to the much weaker interlayer bonding than the intralayer ones.”

12. The word ‘collapse’ is only used in the abstract when referring to the out-of-plane transverse phonon mode. In the subsequent discussion (line 174), the manuscript states that this mode ‘seems to be missing in the experimental data.’ Removal of this mode from the calculation of the heat capacity seems to help reconciling experimental data and computational predictions, which is encouraging – Fig. 4b and lines 241-243. I found the physical arguments put forth to explain these observations quite confusing to follow – i.e., where does this mode (presumably propagating?) go when transitioning from the ‘perfect crystal’ to the real material? Physically speaking, ‘disappearance’ does not seem to be the right term to use, given the general (normalization) properties of the underlying density of states.

Response: We thank the reviewer for this comment and agree with the reviewer, the words “collapse” and “disappearance” are not the right terms. The ZA mode is strongly overdamped due to the slip effect of this van der Waals crystal. We have revised the manuscript and replaced the words of “collapse” and “disappearance” by “overdamped by slip”.

13. In line with the above, the abstract also refers to a ‘frequency resonance’ effect

which does not seem to be explained in any depth in the subsequent discussion.

Response: We thank the referee for the comment. We have added the explanation of this effect in the revised manuscript.

“Meanwhile, dense, low-energy optical modes cross over with each other and are strongly correlated with acoustic modes in the whole Brillouin Zone, particularly, the modes below 5 meV. The pronounced phonon energy/frequency overlap induces the intensive resonant scattering or frequency resonance effect.^{17,18}”

In the abstract, the “frequency resonance” is also revised as “a large acoustic-optical frequency resonance” for better readability.

14. Throughout the manuscript, there is a limited description of the possible role of other defects on experimental observables, particularly those of a thermophysical nature at low temperatures. This point should be addressed in more depth, particularly when singling out interlayer slip as the only mechanism at play in this material.

Response: We thank the referee for the comment. The interlayer slip will introduce some defects, e.g., the stacking faults that have been observed in this work (see Figure R5) and our previous study (Science 369, 542 (2020)). The dense stacking faults are likely to be the dominant, intrinsic defects in InSe crystals.

Apart from the stacking faults directly related to the interlayer slip, we cannot exclude other defects like dislocations, ripplocations, and point defects. However, these defects are not as dense as the stacking faults. Therefore, considering the scope of this study, we focus on the interlayer slip and the resultant stacking faults in InSe.

The main idea of the above discussion has been included in the revised manuscript. “Although we cannot exclude other defects like dislocations, ripplocations, and point defects, the interlayer-slip-induced stacking faults should be the dominant.”

Figure R5 (a) Typical stacking faults and (b)(c) dislocation defects in InSe crystals.

15. Description of the calculations, in the main text and in the Supplementary Material: I found the description of these quite confusing and fragmented all throughout – it needs to be improved considerably. The authors certainly recognize the value of carrying out both harmonic lattice-dynamics calculations as well as AIMD, yet they do not seem to capitalize from their complementarity. To note:

- Supplementary Info, Fig S3: have these been obtained from harmonic calculations? Under the quasi-harmonic approximation or by relaxing unit cell to its internal-energy minimum? Clarify.

Response: We thank the referee for the comment. The original version of Fig. S3 was obtained from DFT harmonic calculations by relaxing unit cell to its internal-energy minimum, rather than under the quasi-harmonic approximation. In the updated manuscript, the Fig. S3 has been revised and the DFT calculation has been removed. We retained the AIMD results, recalculated with higher precision. The lattice parameters were optimized by the NPT in the molecular dynamics.

- Supplementary Info, Fig S4: how do these AIMD calculations compare to those of Fig. S3? A comparison at the level of the phonon DOS (possibly including various projections) should already be telling you something about anharmonic effects, for example.

Response: We thank the referee for the comment. As mentioned in the revised manuscript, all the ab initio results have been recalculated with higher precision, as suggested by the reviewers. Compared in Figure R6, the harmonic phonon at 0 K, both the dispersion and projected DOS, are not far from the AIMD phonons at 200 K. The small discrepancy could be attributed to the slightly different lattice parameters (AIMD results was optimized through NPT) and the methods to extract the 2nd order force constants.

Figure R6 (a) The DFT (red line) and the AIMD 200 K (blue line) phonon dispersions

for InSe. (b) Projected phonon density of states from DFT calculations. (c) Projected phonon density of states from AIMD calculations at 200 K.

Thus, for the consistent purpose, we only retained the AIMD results in the revised manuscript. All the results relating to the 0 K calculations in the previous manuscript, including the phonon dispersion and phonon lifetime, have been replaced by the AIMD result. The original Fig. S3 has been removed in the revised manuscript.

16. Throughout the text: terms and expressions such as ‘giant anharmonicity’ or ‘collapse of the out-of-plane transverse phonon mode’ would need to be put on more quantitative grounds for them to be meaningful – e.g., ‘giant anharmonicity’ compared to what?

Response: We thank the referee for this suggestion and agree that these words need to be revised. Compared to harmonic levels, there are several phenomena implying strongly overdamped characteristics in InSe. 1) In Fig.3, the constant-q cut of LEB phonon band shows only one acoustic phonon branch instead of two acoustic phonons as TA and ZA, indicating the strongly damped (or disappearing) ZA mode; 2) in Fig.4(b), the fitted C_p without ZA contribution agrees with C_p measurement. 3) in Fig.4(c) the lifetime τ_{exp} falls below the IR limit for the low-energy phonons but exceeds this limit for high-energy phonons.

We agree with the view of the reviewer that the term "giant anharmonicity" or "collapse of out of plane transverse phonon modes" may not be suitable for describing the physics. We have changed “giant anharmonicity” into “strongly overdamped” or “large anharmonicity” in the updated manuscript and delete the phrase of “collapse of the out-of-plane transverse phonon mode”. The details of the changes are as follows:

1) “Hence, the phonon energy is comparable with or even smaller than its linewidth because of the giant anharmonicity” has been changed into “Since the phonon linewidths of InSe have the similar temperature- and q-behaviors, the phonons should be strongly damped with the large anharmonicity.” .

2) “This resonates with the idea that giant anharmonicity, and strongly overdamped dynamical modes, can induce glass-like properties without necessarily following from 3D structural disorder.” has been changed into “This resonates with the idea that strongly overdamped dynamical modes can induce glass-like properties without

necessarily following from 3D structural disorder.”

3) “The giant phonon anharmonicity, together with the low-energy and dense phonon dispersion signature, leads to a glass-like boson peak in heat capacity.” has been changed into “The strongly overdamped phonons, together with the low-energy and dense phonon dispersion signature, leads to a Boson-peak-like signal in heat capacity.”.

Reviewer #2 (Remarks to the Author):

The authors performed diffraction, diffuse scattering, and inelastic measurements on vdW crystal samples of InSe, which is a material with exceptional plasticity, using electron diffraction and neutron scattering. The measurements of phonon dispersion, heat capacity, phonon lifetime, and thermal transport were compared with ab initio molecular dynamics simulations. They reveal some interesting physics, in particular, the relation between layer slip and the phonon anharmonicity. The manuscript will be suitable for publication if the authors address the following issues.

Response: We are grateful to the Reviewer for the recognition of interesting physics in the relation between layer slip and the phonon anharmonicity. We have revised the original manuscript in light of the useful comments and suggestions. Below are our detailed responses to each of the comments.

1. The rods along the 001 direction in the diffuse scattering suggests the loss of long-range order. On the other hand, the explanation on the relative slip in the reciprocal space is confusing. Such “slip” indicates correlation between the slips in real space. Does this indicate that the overall lattice takes a different lattice structure from 2H?

Response: We thank the referee for the comment. For the calculation, the slab model is set as 12 layers with the top 6 layers slipping along the (001) plane when calculating the GSFE. For the real crystals, we find that the slip occurs between adjacent “domains (a region consisting of many layers) rather than each single layer. The 2H symmetry does not change. This is just what we observe in Fig. 1e: the 2H symmetry is maintained but the rod is diffusive along c-axis.

2. The missing ZA mode in Fig. 2 needs better labeling. It is difficult to be identified in the existing plots. The difference between the experiment and simulation is not very clear.

Response: We thank the referee for the suggestion. In the updated Fig.2, the calculated ZA mode is plotted as dashed-lines (Figure R7).

Figure R7. The phonon dispersions observed in the dynamical susceptibility $\chi''(Q, E)$ from (a),(b) inelastic neutron scattering measurements at 200 K. The dash-lines are the simulated ZA phonon dispersions by the *ab initio* molecular dynamics simulations.

3. It would be nice to compare the optical phonon eigenvectors with the slip directions.

Response: We thank the referee for the suggestion. We have added corresponding top-view in addition to the side-view subplots in Fig. 2(f). As shown in Figure R8, the shear optical phonon in LEB does vibrate along the slip direction $[1-10]$ (specifically, $1/3[120]+1/3[\bar{1}10]$). This figure is also incorporated in the revised Fig. 4f.

Figure R8(also Fig. 2(f)). Illustration of the four optical phonon modes at Γ .

4. Why is the localization of the optical modes not observed by the inelastic neutron scattering? They shall show as dispersionless signal through the Brillouin zones.

Response: We thank the referee for this comment. We have measured the whole phonon spectra by the inelastic neutron scattering, the dispersionless signal through the Brillouin zones is above 12 meV (Figure R9). In the manuscript, we focused on the acoustic phonons and low-energy optical phonons, so we only reported the low energy phonons. In the revised version, we have included the whole phonon spectra in Fig.S4(c) (SI) and the localization of the optical modes is observed. Due to the low energy resolution, the splitting of LEB and HEB is hard to be observed in the full-energy spectra.

Figure R9. The whole phonon dispersions observed in the dynamical susceptibility $\chi''(Q, E)$ from (a), (b) inelastic neutron scattering measurements at 200 K.

5. Is the instrument resolution subtracted from the linewidth?

Response: We thank the referee for the comment. It is important to subtract the instrument resolution from the measured linewidth, but it is very challenge for a time-of-flight instrument. We used the software Tobyfit to subtract the linewidth.

6. One of the most important observations of the work is the deviation of the C_p from the Debye model and the Boson peak (they are the same feature and shall not be discussed differently). The other important observation is the loss of the ZA phonon model. The manuscript shall focus more on this.

Response: We thank the review's suggestion. We have revised the manuscript and combined the discussion of the C_p from the Debye model and the Boson peak. The observation on the lost ZA is based on 1) In Fig.3, the constant-q cut of LEB phonon band shows only one acoustic phonon branch instead of two acoustic phonons as TA and ZA, indicating the strongly damped (or disappearing) ZA mode; 2) in Fig.4(b), the

fitted C_p without ZA contribution agrees with C_p measurement. 3) in Fig.4(c) the lifetime τ_{exp} falls below the IR limit for the low-energy phonons but exceeds this limit for high-energy phonons. These might not be enough to conclude the absence of the ZA mode, but the overdamped nature of ZA should be confirmed. Hence, we have revised the manuscript and discussed the overdamped ZA more.

Reviewer #3 (Remarks to the Author):

The present work is reporting the lattice dynamics properties of beta-InSe from inelastic neutron scattering (INS), heat capacity and thermal conductivity experiments on single-crystalline samples and ab initio calculations and its connection with interlayer slip characterizing with electron and neutron diffraction experiments.

Notably, the INS experiments show very large phonon dispersion for the two low energy optical phonon branches between 2 and 6 meV and between 5 and 8 meV with significant anharmonicity close to Brillouin zone center. These results show unusual vibrational dynamics linked with interlayer slip, which is quite unique and of high interest for the field of the thermoelectric materials and of 2D van der Waals layered chalcogenides. These interesting results are analyzed with the help of ab initio calculations but they must be very much improved as there are many uncertainties with them as will be discussed and there are not able to explain the results of thermal conductivity. Also, the discussion on the heat capacity seems unsatisfactory.

Therefore, if the INS experiments are of high interest, potentially publishable in Nature Communications, the present work is still far from the high standard of Nature Communication. However, I am convinced that the present work could perhaps be published in Nature Communication after a major revision following the comments detailed below.

Response: We are grateful to the Reviewer for the recognition of high interest on the research. We have revised the original manuscript in light of the useful comments and suggestions. Below are our detailed responses to each of the comments.

1) Structure characterization

1. The authors did not give the crystal structure obtained from their diffraction data. They must do it.

Response: Currently, we have the data of electrons diffraction (Fig. 1b) and neutron elastic diffraction (Fig. 1c). The two sets of data are consistent with each other, indicating a 2H structure rather than 3R. In addition, the single crystal X-ray diffraction has been conducted and added to further distinguish β -InSe (P6(3)/mmc) with ϵ -InSe (P-6m2) that are both 2H structures. As shown in Fig. R10, the InSe specimen is probably the β -phase due to lower R_{int} values (0.11 for β -phase compared to 0.25 for ϵ -phase). This information has also included in the revised manuscript and SI. Regarding

the two phases, please check more details in the reply to the next comment.

Figure R10(also Fig. S1) Comparison between the observed Bragg reflection intensities at 1.7 K on the single-crystal XRD and the simulated ones based on the β -phase structure. The solid line is a guide to the eye.

2. Also, the authors did not inform the reader of the existence of the epsilon phase with the same lattice parameters than the beta phase but with slightly different crystal structure, they must do it. I don't think that the use of Raman scattering for distinguishing between the beta and epsilon phases is enough.

Response: Thanks for reminding this point. Yes, both β - and ϵ -phases belong to the 2H structure. After re-analyzing our data and conducting new experiments, we find the crystal in this work is probably the β phase based on the following evidence.

i) XRD refinements favor the $P6_3/mmc$ (β -phase) due to lower R_{int} values. Please see the details in our response to the comment #1.

ii) As shown in Fig. 1b and 1c, the Bragg reflections do not move in **ab**-plane, indicating that the phase cannot be γ -InSe. The space group of β -InSe is $P6_3/mmc$ (No. 194), and the space group of epsilon-InSe is $P-6m2$ (No. 187). According to Fig.1d, there are no $l = 2n + 1$ Bragg reflections in (HHL) and (K-KL) plane, and the spindle-like diffuse signal between Bragg reflections along the c^* -axis is observed, which present the correlation between the layers and there is no shifting. All these features point towards the β - rather than the ϵ -phase. This discussion has been added to the manuscript.

iii) The Raman peak around 199 cm^{-1} can be useful to distinguish between the two

phases according to previous reports based on group theory. In the references of Solid State Commun. 25, 5 (1978); Solid State Commun. 29, 31 (1979); Phys. Stat. Sol. (b) 103, 123 (1981), the Raman modes for InSe were deduced by the standard group theory. The β -InSe belongs to the P63/mmc space group with the inversion center. Among the 24 normal modes at the center of the Brillouin zone, 6 are Raman active. The ϵ -InSe belongs to the P-6m2 space group without central symmetry. There are 11 Raman active modes of its 24 normal modes. Among that, the character result in that the A_1 -LO vibrational mode only appears in the ϵ -phase, but not in the β -phase. The frequency of this mode is proposed around 199 cm^{-1} (Solid State Commun. 25, 5 (1978)).

3. I think that additional calculations are needed for confirming that. See the calculations for more details.

Response: We thank the referee for the comment. The additional calculation has been applied in the revised manuscript.

2) Inelastic neutron scattering experiments

4. The authors have analyzed their data using gaussian functions. However, giving the broad linewidth of the observed lines, it seems that the damped oscillator model should be better adapted and therefore to use lorentzian functions instead of gaussian functions.

Response: We thank the referee for the comment. It is important to deconvoluted the instrument resolution from the measured linewidth, and we used the software Tobyfit to obtain the intrinsic linewidth from the observed lines. The old version of Tobyfit only includes Gaussian function, which had the similar q-dependence of the Lorentzian function with elastic peak subtraction. We have revised code of Tobyfit, and applied the Lorentzian function to fit the phonon linewidth in the revised manuscript. The Figure 3 and Figure 4 have been updated.

5. I think that the HEB is also interesting and the energy dispersion must be shown as for the LEB in Fig. 3(a) and (b), at least in the supplementary information. Concerning the linewidth of such HEB, as it seems to correspond to two different modes, it is maybe not needed to show them, but this could be still interesting to show them.

Response: We thank the referee for this suggestion. Compared to LEB with TA and TO, HEB is very complicated with the compositions of LA, LO and TO. As the manuscript

focus on the anharmonicity of LEB, we didn't include it in the manuscript. We have included the linewidth of HEB in SI of the revised manuscript (Figure R11, also Fig. S6). The HEB phonon dispersions are not temperature dependent. Although the phonon linewidths of HEB are smaller than the values of LEB, the dispersion of HEB is flatter than the phonon of LEB and the group velocity is small. Hence, the lattice thermal conductivity of HEB is small.

Figure R11(also Fig. S6) The fitted HEB phonon dispersions and linewidths along GM (a, c) and GK (b, d).

6. The discussion in lines 188-191 must be revised because it is inaccurate. Indeed, the authors must take more care when comparing with SnSe and PnTe/SnTe. Indeed, in SnSe the phonon dispersion is similar in the Gamma-Y direction but not in the other directions and it experienced phase transition at 800 K (see ref. 14). Despite that, the energy of low energy TO mode decreases with increasing T, which is not the behavior of a soft mode. On the other hand, the TO mode of SnTe and PbTe have a soft mode behaviour with their energies increasing with increasing T (refs. 15). This is a different behavior than in InSe, where it has more normal behavior with energies decreasing with increasing T. Therefore, the discussion in lines 188-191 must be modified.

Response: We thank the referee for these comments. Since SnSe, PnTe and SnTe are typical IV-VI group materials, we tried to explain that different scattering mechanisms have different phonon linewidths/group velocities by comparing them. We are sorry for the inaccurate description. The discussion has been modified:

“Unlike the 2D-like scattering of SnSe^{19,20} and nano-domain effect of AgSnTe₂²¹, InSe behaves as the “waterfall” effect in PbTe²² and “nesting” effect in SnTe¹⁴: the excitation mode “1” TO mode at Γ ($q=0$) is broad, even down to 0 meV, which originates from the weak vdW force between the layers and the strong TA-TO interaction.”

7. In the figure 3 (c), the authors have added the q variation of the linewidth of SnTe, PbTe and SnSe from the literature. There are some problems with these data. For PbTe, in ref. 16 (also in ref. 19), the data correspond to acoustical phonons and not optical mode and therefore this is nonsense to compare them with the results obtained for InSe. For SnSe, the data in ref. 17 correspond to Na doped SnSe and not pure SnSe and it is not clear how the data plotted in fig. 3(c) have been obtained. For SnTe, the data in ref. 18 looks to have broader Q range.

Response: We thank the referee for this suggestion. The data of PbTe, SnTe, and Na-doped SnSe have been removed. For SnTe, the data were obtained from Ref. [18] and there were only two data points at $q=0.1$ and 0.125 along [110] direction at 200K, hence, we also removed the data of SnTe from Fig.3(c) and only briefly discussed it in the text. “All phonon modes are broad (1~3meV) comparable to the typical binary thermoelectric alloys with the strong phonon-lattice or phonon-phonon interaction as PbTe and SnTe.^{14, 23} The phonon linewidth (at Γ -point) of PbTe (incipient ferroelectric phonon-lattice interaction) increases with increasing temperature²², while a decrease of the TO linewidths is observed with increasing q for SnTe^{14,23}. Since the phonon linewidths of InSe have the similar temperature- and q -behaviors, the phonons should be strongly damped with the large anharmonicity. ”

8. Note that in the Methods part, it is incorrectly written that the INS experiments were performed at 220 K instead of 200 K.

Response: We thank that the referee helps us point out the typo. The temperature has been corrected as 200 K.

3) Heat capacity and thermal conductivity

9. I don't agree with the use of Boson peak for the Debye plot of the heat capacity because it means that it should be linked with disorder and/or glassy behavior. In fact, the Debye model can be applied strictly on the case with only one atom where there is only acoustical phonon. In more complex case, there is always deviation from the

Debye behavior because of the presence of optical modes. When there is low energy optical modes, it is well documented that there is always a peak in the Debye plot of the heat capacity both in experiments and in calculations. Therefore, the authors must remove all the paragraph from the lines 256 and 279.

Response: We thank the referee for this suggestion, and agree that the anomaly of C_p/T^3 may be not exclusively an exact Boson peak. Comparing the Boson peak with our data, they have the similar macroscopic consequences while the specific microscopic origin can be different from the original context of glasses where structural disorder plays a fundamental role for the Boson-peak. In the manuscript, we have replaced the term of “Boson peak” with “Boson-peak-like”. Meanwhile, we tried to discuss this broaden signal and want to introduce some background in this paragraph and revised the manuscript. The paragraph from the lines 256 and 279 has been revised.

“An anomalous hump of C_p/T^3 is observed at around 12 K, Fig. 4(b), indicating a deviation from Debye law, $C_p \sim T^3$. Although similar behavior could be observed in a large plethora of crystalline materials such as thermoelectrics,²⁴⁻²⁶ incommensurate structures,^{27,28} superionic conductors,²⁹ orientationally disordered crystals,³⁰ ferro-elastic memory alloys,³¹ metal halide perovskites,³² ferroelectric materials,³³ organic materials and even molecular crystals without any clear sign of disorder³⁴, the fundamental origin of this phenomenon is still a subject of strong debate and several theoretical models have been proposed.³⁵⁻⁴¹ The thermal properties in these crystalline materials are evidently not a direct manifestation of positional disorder⁴²⁻⁴⁴, for example glasses^{45,46}, but they are commonly related to a softening of acoustic-optical modes or to strong interactions between low-energy optical modes and the acoustic modes, often as a result of enhanced anharmonicity.^{29,47-51} As of now, the universal character of these Boson-peak-like features is still under scrutiny.⁵² Partial glassy behavior in crystalline or weakly-disordered systems seems to be a concrete possibility, hinting towards possible different origins for the different glassy features. We will discuss the case of InSe within this general picture.”

10. There is also generally some deviations between the calculations and the experiments. It is not surprising that there is some deviations in their case, notably

because the harmonic calculation did not take into account to the large anharmonicity of the LEB and HEB and also maybe because of the disappearance of the ZA branch because of the large disorder in the c direction, as proposed by the authors. Because of the large dispersion of the LEB and HEB, the Einstein model is not valid anymore. More sophisticated model including both the large dispersion of the HEB and LEB must be used. However, I think that it is just simpler to remove the part on the Debye-Einstein model fitting the heat capacity data.

Response: We thank the referee for this suggestion. Following your kind comment, in the revised manuscript, we instead employ the more sophisticated AIMD technique to calculate the lattice dynamics and fit the heat capacity (see Fig. 4a and 4b).

As to the Debye-Einstein model, although it is a simple and preliminary model as compared to AIMD, it is still a classical, intuitive and widely used model to analyze the heat capacity. Given the popularity of this model, we still leave it in SI as the supplemental evidence in discussing the partial density-of-state of phonons for the strong acoustic-optical phonon interaction.

11. I don't agree with the authors that there is flattening in their thermal conductivity data. Indeed, the thermal conductivity must be plotted in a linear scale, otherwise it is more difficult to see the decrease of the thermal conductivity with increasing temperature with semi-log scale used in Fig. 4 (d). For the out-of plane thermal conductivity, one can see that it decreases from about 2 W/m.K at about 50 K down to about 0.8 W/m.K whereas in the case of the in-plan thermal conductivity, which is rather large, it decreases from about 10 W/m.K at about 50 K to 6 W/m.K at 300 K. Therefore is maxima in the thermal conductivity, even in the case of out of plane thermal conductivity. Therefore this give some doubts about the glassy behaviour claimed by the authors. Do the authors check if the thermal conductivity is following $1/T$ dependence ?

Response: We thank the referee for this suggestion. We have changed Fig4(d) with a linear scale. The thermal conductivity follows $1/T$ dependence above 150 K (Figure R12). This T^{-1} behavior has been included in the revised manuscript.

Figure R12(also Fig. 4(d)) In-plane (black squares) and out-of-plane (blue circles) thermal conductivity. The inset shows the low temperature thermal conductivity with logarithmic scale.

12. Also, I think that in the inset of Fig. 4 (d) in log-log scale, they must show the power laws followed by thermal conductivity. Concerning the discussion on the thermal conductivity, I think that it must ne mode discussed in light of the DFT calculations. See the comments on calculations below.

Response: We thank the referee for this suggestion. For the inset of Fig. 4 (d) in log-log scale, we presented κ_L -T relation. Unlike the T^3 for a crystal and T^2 for glasses, the powers of InSe are 1.6 and 1.4 or the in-plane and out-of-plane cases, respectively, which suggested complicated scattering processes. The discussion has been revised as “Typically for a crystal, boundary scattering is the dominant scattering mechanism at low temperature and κ_L roughly follows the T^3 scaling. Meanwhile, a T^2 dependence is usually observed for glasses.⁵⁸ For the inset of Fig. 4(d), the κ_L -T relation significantly deviates from the T^3 law below 50 K, instead being $T^{1.6}$ and $T^{1.4}$ for the in-plane and out-of-plane cases, respectively. This deviation suggests more complicated scattering processes of InSe.”

4) Calculations

13. I don't think that the calculations are sufficiently converged. Indeed, for the structural relaxation of the primitive cell, the authors used a 4x4x3 k-point mesh despite the lattice parameter a is about 4 angstroems and the lattice parameter c is about 17 angstroems. Clearly, the density of k points is to small in the a and b directions. I would expect to use 12x12x3 k-point mesh. Also, the use of 1x1x1 k-point mesh for the 4x4x1

supercell is certainly insufficient as well. Instead, we would expect to use 3x3x3 k-point mesh. Therefore, the authors must redo their calculations with finer k-point mesh. They must also indicate if they use Gamma centered or Monkhorst pack grids. They must give the criterion used for the force convergence. They must compare their calculated crystal structure with the experimental one obtained from their diffraction data and with the literature (in the supplementary information).

Response: We thank the referee for the comment. Following the insightful suggestion

and considering the data consistency, we have switched entirely to using the results of

A

I

M

D

(

u

14. Concerning the phonon dispersion curves, they must show them not only along Gamma-K and Gamma-M but also in the Gamma-A, Gamma-L and Gamma-H directions. This will give very useful informations. They must also give the list of the phonon modes at Gamma point with their symmetries and energies, as Bejani et al (Phys. Rev. Mater. 6, 115201 (2022)) and compare them with their experimental results as well as the literature results (Raman and infrared spectroscopy) in the supplementary information. They must also give the symetries and the energies of the three optical modes in Figure 2(f). As Bejani et al have performed some previous lattice dynamics calculations on beta-FeSe with DFT, they must be cited and their results must be compared with the present results.

Response: We thank the referee for the comment. Following the insightful suggestion

and considering the data consistency, we have switched entirely to using the results of

and in the following table, we labeled the symmetries and energies of all the optical phonons at Γ point. The modes with Raman / infrared active are also

labeled in the table. The E_{2g}^1 , $B_{1g}^1(2)$, $E_{2u}^1(3)$, and $E_{1g}^1(4)$ modes correspond to 1-4 optical modes in Figure 2(f), respectively. Compared with the results in Phys. Rev.

Mater. 6, 115201 (2022), Table II, the symmetries are the same, and the phonon

e

d

a

frequencies are slightly different due to the different exchange-correlation functionals (R2SCAN metaGGA in the work vs. LDA), lattice parameters, and the methods to compute force constants (AIMD in this work vs. DFPT). The reference of Physical Review Materials 6, 115201 (2022) has been cited in the revised manuscript.

Furthermore, the frequencies of the Raman active modes are also consistent with the experimental measurements in this work (Fig. S1 and S4 in the original SI), except for the E_{2g} mode. The missing of the E_{2g} mode in the Raman measurement of β -InSe was previously discussed in Phys. Stat. Sol. (b) 103, 123 (1981).

Figure R13(also Fig. S4 (a), (b)) The phonon dispersions with(a) and without(b) LO-TO splitting considered from AIMD simulations at 50 K, 200 K, and 400 K.

Table R1(also Table S1) The frequencies, symmetries, and Raman/IR activeness of the Γ -point phonons for β -InSe at 200 K. The numbers in brackets correspond to four optical phonon modes in Fig. 2f.

β -InSe AIMD 200K							
meV	2.03	3.78	4.71	5.01	12.90	13.70	21.66
cm ⁻¹	16.4	30.5	38.0	40.4	104.0	110.5	174.7
Irre. Rep.	E_{2g}^1 ① in Fig.2f	B_{2g}^1 ② in Fig.2f	E_{2u}^1 ③ in Fig.2f	E_{1g}^1 ④ in Fig.2f	B_{1u}^1	A_{1g}^1	E_{1g}^2
Raman active	√	-	-	√	-	√	√
meV	21.75	22.73	22.90	24.06	24.41	28.49	28.53
cm ⁻¹	175.4	183.3	187.7	194.0	196.9	229.8	230.1
Irre. Rep.	E_{2u}^2	E_{2g}^2	E_{1u}	A_{2u}	B_{2g}^2	B_{1u}^2	A_{1g}^2
Raman active		√	-	-	-	-	√

15. Giving the importance of the LO modes for distinguishing with Raman scattering experiments between the beta phase and the gamma and epsilon phases, and more particularly the last one, it is essential to also calculate the LO-TO splitting. Concerning the distinction between the beta and the epsilon phases, the authors must indicate more clearly that there is a claim that beta and epsilon phases can be distinguished with the absence or presence of the Raman line at about 199 cm⁻¹ which should a LO mode of the epsilon phase because the LO modes are both Raman and IR active in the epsilon phase but there was no theoretical support of this claim. Therefore, harmonic lattice dynamics calculations with DFT including LO-TO splitting must be performed for confirming this claim. This is important because everything distinguishing the beta and epsilon phases is based on that claim that have never really checked with calculations. We must be sure that the present sample has beta structure and not epsilon structure.

Response: We thank the Reviewer's comment. The samples can be indexed to the β -phase rather than the ϵ -phase according to the single-crystal XRD data (Fig. R10/S1) and the neutron diffuse scattering data (Fig. 1(d)). Please check our response to the comment #1 and #2.

As shown in FigureS14, the AIMD calculated phonon dispersions of β -InSe with and without LO-TO splitting are given at 50 K, 200 K, and 400 K. The Raman active modes are also summarized in Table R1. The frequencies of the Raman active modes are also consistent with the experimental measurements except for the E_{2g} mode. The missing E_{2g} mode in the Raman measurement of β -InSe was previously discussed (Phys. Status Solidi B, 1981, 103, 123-130).

Regarding the different Raman modes, the authors deduced the Raman modes for InSe based on the standard group theory (Solid State Commun. 25, 5 (1978); Solid State Commun. 29, 31 (1979); Phys. Stat. Sol. (b) 103, 123 (1981)). The β -InSe belongs to the P63/mmc space group with the inversion center. Among the 24 normal modes at the center of the Brillouin zone, 6 are Raman active. The ϵ -InSe belongs to the P-6m2 space group with non-central symmetry. There are 11 Raman active modes of its 24 normal modes. Among that, the character result in that the A₁-LO vibrational mode only appears in the ϵ phase, but not in the β phase. The frequency of this mode is proposed around 199 cm⁻¹ (Solid State Commun., 1978, 25, 5).

In this work, our present data (single-crystal XRD, neutron diffuse scattering) have been enough to support the β -InSe. In view of the scope of this study, as to the Raman

issue, we simply follow the arguments by previous reports, and more detailed investigation can be done in the future.

16. The phonon lifetime must be as well calculated with finer k-point mesh.

Response: Thank the Reviewer for the suggestion. The phonon lifetimes, which was obtained from DFT calculations, have been replaced by the AIMD 50 K and 200 K results in the revised manuscript (Fig. 4c), also as shown below (Figure R14). Generally speaking, the lifetime at 50 K is higher than that of 200 K, indicating a weaker phonon-phonon interaction at lower temperatures.

Figure R14 The temperature-dependence of Energy vs. phonon lifetimes by theoretical the simulation.

17. It is clear for me that the very large anisotropy of the experimental thermal conductivity must better analyzed using notably DFT calculations. For a better analysis and discussion of the results of thermal conductivity, the authors must calculate the thermal conductivity from all the previous calculations they have done and also plot the group velocities as function of the energies and/or modes.

Response: Thank the Reviewer for the suggestion. We've calculated the group velocity vs. energy and the accumulated thermal conductivity along the two directions, as shown in Figure R15 below. As the referee has pointed out, there is indeed strong anisotropy in terms of phonon group velocities. The in-plane group velocities are several times larger than out-of-plane ones starting from 3 meV. The anisotropy is also reflected in the cumulative lattice thermal conductivities. Furthermore, the referee's comment is

also correct and predictive that the majority of the in-plane lattice thermal conductivity is contributed from the phonons between from 2 meV to 10 meV while the out-of-plane lattice thermal conductivity is almost saturated below 3 meV.

18. Indeed, I don't agree with their interpretation of the thermal conductivity. Clearly, the LEB and HEB have quite large dispersion and we can expect that their group velocities must be high enough for having a large contribution to the thermal conductivity. For better seeing that, a calculation of the cumulated thermal conductivity vs energy is needed for both a and c directions and probably also from the different phonon branches. I am convinced that one of the main reasons of the anisotropy of the thermal conductivity (but not the only one, certainly the disorder play also one important role) is the largest contribution from the optical phonons below 9 meV.

Response: We addressed this and the above points together. We have calculated the group velocities and cumulative lattice thermal conductivities, both from the AIMD 200 K, as shown in Figure R15. As the referee has pointed out, there is indeed strong anisotropy in terms of phonon group velocities. The in-plane group velocities are several times larger than out-of-plane ones starting from 3 meV. The anisotropy is also reflected in the cumulative lattice thermal conductivities. Furthermore, the referee's comment is also correct and predictive that the majority of the in-plane lattice thermal conductivity is contributed from the phonons between from 2 meV to 10 meV while the out-of-plane lattice thermal conductivity is almost saturated below 3 meV.

Figure R15 (a) Group velocity in plane and out of plane for InSe as the function of energy. (b) Cumulative lattice thermal conductivities at 200 K as the function of energy.

19. Concerning the AIMD calculations, the phonon dispersion curves are shown in Fig.

S4 (a) only below 16 meV. Please show the data for the full energy scale and compared the 50 K data with the data from harmonic calculations.

Response: Thank the Reviewer for the suggestion. In the revised version, we only included the whole phonon spectra by AIMD in the revised manuscript. According to the suggestion of the reviewer, we also compared the calculation results of AIMD and DFT, and they are similar in Figure R16.

Figure R16 (a) Temperature-dependent phonon dispersion by AIMD simulations with LO-TO splitting. (b) The DFT phonon dispersions for InSe.

20. Indeed, at 50 K the phonon dispersion from AIMD look very different from the harmonic phonon dispersion, especially the branches between 10 and 16 meV, especially when approaching the Brillouin zone center for which the energy is much smaller in the AIMD calculations than in the harmonic calculations. Even for the lower HEB and LEB, the energies close to Gamma point in the AIMD are significantly larger than in the harmonic calculations. Have you some explanation for these strong disagreements? I suspect again that this is linked with the insufficient convergence conditions.

Response: Thank the Reviewer for the comments and suggestion. We have replaced all the harmonic calculations (DFT) by AIMD in the revised manuscript for consistency. We also reoptimized the unit cells with the parameters suggested by the referee, and the results of AIMD calculations are more systematic, as will be discussed in the next point. Besides, though the harmonic results will not be shown in the revised manuscript, we have compared the harmonic ones with AIMD ones (Figure R17). The small discrepancy between the DFT and AIMD results could be attributed to the slightly different lattice parameters (AIMD results was optimized through NPT) and the

methods to extract the 2nd order force constants.

Figure R17 (a) The DFT (red line) and the AIMD 200 K (blue line) phonon dispersions for InSe.

21. Strangely, the calculated Raman spectra in Fig. S4 (b) have low energy mode at 2-2.5 meV but this does not correspond to the energies of the lowest energy mode in the phonon dispersion curves in the Fig. S4 (b). It is also not clear to which phonon mode in the phonon dispersion curves is corresponding the Raman mode at 5 meV. Could you explain this disagreement? Here, like for the harmonic modes, it is necessary to well identify the different vibrational modes and their symmetries.

Response: In the revised manuscript, we have recalculated AIMD results with different parameters (Figure R16(a)). Though the E_{2g}^1 mode at 50 K is overestimated, at around 3.41 meV, it decreased down to 2.03 meV at 200 K, and 1.92 meV at 400 K. The decreasing trend of this mode, as long as that of the E_{1g}^1 , are captured, and consistent with our Raman measurement at low frequency range (Fig. S4b in the original manuscript), and the measurement of energy shift of LEB (Fig. 3).

22. Also, it is necessary to give the method used for calculating the Raman spectra.

Response: We thank the reviewer for this comment and sorry for confusing the reviewer. The Raman spectra (Fig. S2 and Fig. S4) are the *experimental* data. In order to void the unclear statement, we have labeled the Raman spectra in the caption and text.

“The temperature-dependent Raman spectra was performed in a homebuilt closed-cycle optical cryostat down to 1.6 K. A He/Ne laser centered at 633.1 nm was employed as the excitation source. A combination of one reflective Bragg grating and two Bragg notch filters allowed measurements down to $\sim 5 \text{ cm}^{-1}$. The laser power was kept below

50 μW to prevent significant laser heating of the samples.”

REVIEWER COMMENTS

Reviewer #1 (Remarks to the Author):

I have read in detail the (quite extensive) revision of the original submission, and I am not in a position to recommend this work for publication. My primary concern revolves around the claims (in my view still unsubstantiated) of the unambiguous existence of low-energy overdamped phonons in this particular material. The following comments should be sufficient for the authors to understand the reasons why:

- The title continues to be misleading, as it appears to imply that ‘overdamped phonons’ have been identified in this 2D material.
- In the Abstract, the authors state that ‘... the presence of extremely soft optical modes induces a Boson-peak-like feature in the heat capacity and strongly damped lattice thermal conductivity.’ Again, this statement is also misleading, the heat capacity per se shows no anomalies to this effect, just deviations from Debye’s model. To put things in perspective, I quote what has been for (quite some time now) regarded as ‘expected’ and not ‘anomalous’ in the field: ‘... deviations from Einstein and Debye models of specific heats, at one time considered to be anomalous variations of specific heats, are now taken as normal in the light of detailed lattice calculations’ [see E.S.R. Gopal, Specific Heats at Low Temperatures, Plenum Press, New York, 1966, p. 158].
- The above (and quite categorical) assertions are not followed up throughout the manuscript in a satisfactory manner. For example, in Line 179, it is stated that ‘this phonon mode ... is likely overdamped ...’ As far as I can judge, the revised manuscript still does not provide sufficient evidence to this effect:
 - o Figure 2: I cannot really see from this figure how (as claimed in, for example, Line 176) ‘... the out-of-plane transverse acoustic branch (denoted as ZA with dash-lines, and similar to that in 2D materials) with the lowest energy seems to be strongly overdamped in the experimental data.’
 - o Figure 3: it is still not clear at all whether the authors have been able to extract reliable linewidth information from the neutron data, which can be related to actual excitation lifetimes. This concern is also in line with the very significant discrepancies between experiment and calculation shown in Fig. 4c, where agreement is not even qualitative. On this basis alone, I cannot understand how the authors can claim that they have solid evidence for their claims.
 - o I also have concerns about the calculations themselves, including how these have been used to compare against experimental data:
 - In their revision, the authors have chosen to focus on the use of AIMD and, for example, simulations at much higher temperatures are employed to calculate the heat capacity all the way down to a few K. I do not understand this choice, yet it also appears to be a key point to support the existence of overdamped modes – see Line 180: ‘More evidence for this overdamped mode will be provided in the analysis of the heat capacity.’ One would think, for example, that a harmonic calculation at zero temperature should be the natural starting point to compare these heat-

capacity data with the calculations, as well as to establish whether it is necessary to invoke departures from this approximation at these very-low temperatures. On the experimental front, inelastic neutron-scattering data at these low temperatures would also have been the best way to compare reality (experiment) with computational predictions. These data are not reported in this work.

- In a similar vein, the revision falls quite short at explaining how a number of observables (particularly excitation lifetimes) have been calculated, including the level of theory and the approximations involved.

Reviewer #2 (Remarks to the Author):

The authors made a good effort to revise the manuscript extensively. They address most of my concerns and I believe the manuscript is suitable for publication in the current form.

Reviewer #3 (Remarks to the Author):

The authors have revised their work as requested by the different reviewers, which has very much improved their article.

Their work can now be accepted for publication in Nature Communication after correcting small typo error indicated below.

There is still a small typo error in line 206 :

in ref. 21, the compound studied is AgSbTe₂ and not AgSnTe₂ as written in the revised manuscript.

Reviewer #1:

General comments: I have read in detail the (quite extensive) revision of the original submission, and I am not in a position to recommend this work for publication. My primary concern revolves around the claims (in my view still unsubstantiated) of the unambiguous existence of low-energy overdamped phonons in this particular material. The following comments should be sufficient for the authors to understand the reasons why:

Response: We thank the Reviewer for the dedication in assessing our work. The main concern of Reviewer 1 remains the lack of proofs behind the claim that low-energy phonons in InSe, and in particular the ZA phonon, are overdamped. Here, we summarize and emphasize the main reasons that led us to this conclusion.

1) In Fig. 2(a-b), the out-of-plane transverse acoustic branch denoted as ZA in the experimental data deviates from the AIMD calculations and does not clearly appear in the experimental data. This proves that a harmonic approximation (as used in AIMD) cannot correctly describe the phonon dynamics in experiments, and the phonons are strongly anharmonic, or equivalently overdamped.

2) In Fig. 3, the constant-q cut of LEB phonon band shows only one acoustic phonon branch instead of two acoustic phonons as TA and ZA. The absence of a well-defined peak implies that the linewidth of the ZA mode is larger than its characteristic energy, indicating the strongly damped nature (or as denoted in the previous version of our manuscript, “disappearance”) of the ZA mode, consistent with point (1).

3) The overdamped nature of the low-energy phonons is further indicated by their short lifetime τ_{exp} , which falls below the Ioffe Regel limit (Fig. 4c).

4) The measured heat capacity can be reconciled with the calculated one only if the contribution of the ZA mode is neglected. This can be again understood by considering that the ZA mode is very short living and not able to contribute to thermal transport because of its overdamped nature. When the ZA is subtracted, the measured and calculated heat capacities agree well (see Fig.4(b)).

In summary, we have provided four independent reasons that prove the overdamped nature of the low-energy phonons, in particular the ZA mode, in InSe

crystal. We believe that this is enough to substantiate our claim. To improve the clarity of our message, we have added two new sentences regarding this important point in the conclusions.

Comment 1: The title continues to be misleading, as it appears to imply that overdamped phonons have been identified in this 2D material.

Response: In our opinion (please see reply above), we have convincingly proved the overdamped nature of the low-energy phonons, and in particular the ZA mode, in InSe. The phonon spectra is a direct proof for the overdamped phonons and there was no report before, hence, we do not perceive the current title as misleading. We are happy to follow further concrete suggestions for the title from Reviewer 1.

Comment 2: In the Abstract, the authors state that the presence of extremely soft optical modes induces a Boson-peak-like feature in the heat capacity and strongly damped lattice thermal conductivity. Again, this statement is also misleading, the heat capacity per se shows no anomalies to this effect, just deviations from Debye model. To put things in perspective, I quote what has been for (quite some time now) regarded as expected and not anomalous in the field; deviations from Einstein and Debye models of specific heats, at one time considered to be anomalous variations of specific heats, are now taken as normal in the light of detailed lattice calculations [see E.S.R. Gopal, *Specific Heats at Low Temperatures*, Plenum Press, New York, 1966, p. 158].

Response: We thank Reviewer 1 for this comment and for pointing out this interesting reference. We have cited this reference as Ref. [24]. There is indeed some confusion on the usage of the term “anomaly”. We do agree with the first page of chapter 7 in the above-mentioned reference that the word “anomaly” could carry a negative connotation that is certainly not necessary. We would like to explain nevertheless that the word “anomalous” was borrowed from the glasses community where it is universally used to indicate deviations from the Debye paradigm (see for example the title of the recent comprehensive book on the topic, “*Low-Temperature Thermal and Vibrational Properties of Disordered Solids: A Half-Century of Universal “Anomalies” of Glasses*”

by M.A.Ramos, World Scientific 2022).

In our manuscript, the direct evidence of phonon overdamping is the phonon spectra by the inelastic neutron scattering measurements, as explained in our response to the General comments. The deviation of specific heat from the Debye model is only a partial proof of phonon overdamping on the macroscopic manifestation.

To avoid confusion, we have replaced the term “**anamoly**” by other terms as “**deviations from Debye model**” or “**glassy behavior**” as commonly used in this context (Phys. Rev. B 23, 3886, 1981; Phys. Rev. B 99, 024301; and many more). Both the abstract and the discussion on the heat capacity have been revised accordingly, and the new reference indicated by Reviewer 1 has been added.

Comment 3: The above (and quite categorical) assertions are not followed up throughout the manuscript in a satisfactory manner. For example, in Line 179, it is stated that this phonon mode is likely overdamped; As far as I can judge, the revised manuscript still does not provide sufficient evidence to this effect:

Response: Please kindly see our above reply to the General comments on this point.

a) Figure 2: I cannot really see from this figure how (as claimed in, for example, Line 176) the out-of-plane transverse acoustic branch (denoted as ZA with dash-lines, and similar to that in 2D materials) with the lowest energy seems to be strongly overdamped in the experimental data.

Response: As the phonon could be scattered, the experimental data contain information of both phonon dispersion and lifetime. Usually, there are two TA and one LA for for a cubic crystal; for layered-structured InSe, the acoustic phonons are labeled as one TA, one ZA and one LA, where the ZA is defined as the out-of-plane transverse acoustic branch. In Fig.2(a-b), the white lines and white dashline are the AIMD data. We could clearly observe that the ZA branch does not agree with the simulation as its dispersion seems to be absent in the experimental data. This picture is further confirmed (as explained in points 2 and 3 above) by constant-q cuts and the extremely short lifetime

and it is therefore favored. These observations corroborate the overdamped nature of the ZA mode.

b) Figure 3: it is still not clear at all whether the authors have been able to extract reliable linewidth information from the neutron data, which can be related to actual excitation lifetimes. This concern is also in line with the very significant discrepancies between experiment and calculation shown in Fig. 4c, where agreement is not even qualitative. On this basis alone, I cannot understand how the authors can claim that they have solid evidence for their claims.

Response: We have adopted the standard, sophisticated procedure to process the neutron data and extract the information of the linewidth (D.L. Abernathy, et al., EPJ Web of Conferences 83, 03001 (2015); R. A. Ewings, arXiv:1812.08583, etc.). The details are described below. The linewidths were obtained from the full width at half maximum (FWHM) of the experimental INS data, then we can further explore phonon lifetime based on the relationship $\tau(q) = \hbar / \text{FWHM}(q)$. Although the shape of neutron beam from a Spallation-Neutron-Source is hard to be decided, the ISIS Pulsed Neutron and Muon Source, STFC Rutherford Appleton Laboratory, develops a software TOBYFIT to get FWHM by taking into account the broadening of the data arising from the resolution of the instrument for a time-of-flight instrument. More details on the software of TOBYFIT can be found from the manual on the website (<https://www.isis.stfc.ac.uk/Pages/tobyfit-manual6825.pdf>). Regarding this issues, we have detailed the description in the Method session: “Linewidths from INS are extracted by *tobyfit* in Horace taking into account the broadening of the data arising from the resolution of the instrument.”

“The very significant discrepancies between experiment and calculation shown in Fig. 4c” is just our statement; the lifetime τ_{exp} is *not* consistent with the τ_{AIMD} , but being orders of magnitude lower. This is a clear proof that harmonic dynamics cannot capture the behavior of the phonons in InSe, pointing towards the strongly anharmonic, overdamped nature. Together with the falling below the IR limit for the low-energy

phonons, and the heat capacity discrepancy mentioned above, we confirm the overdamped nature of ZA.

c) I also have concerns about the calculations themselves, including how these have been used to compare against experimental data.

Response: Thanks for this question. Regarding the details of the calculations themselves, please check our response to comment 5. The comparison between the calculation and experiment has been clarified in the manuscript. Several key points are summarized below.

(i) The white lines in Fig. 2a-b are the calculated phonon dispersions. Fig. 2c-d are intensity plots considering dynamical structure factor and instrument resolution. They are both calculated by AIMD simulation and captured the features of the spectrum including phonon energies and INS intensities. However, the out-of-plane transverse acoustic branch denoted as ZA in the experimental data deviates from the AIMD calculations and does not clearly appear in the experimental data. This proves that a harmonic approximation (as used in AIMD) cannot correctly describe the phonon dynamics in experiments, which indicates that the phonons are strongly anharmonic, or equivalently overdamped.

(ii) In Fig. 4a, we simulated specific heat C_{AIMD} from the calculated phonon DOS. The experimental and computational data are consistent with each other between 50 K and 200 K. Noticeably, the calculated C_p is larger than the measured one below 50 K, inset of Fig. 4(a). Instead, by excluding the ZA mode in calculation, we can reach a good consistency between the experimental and computational data. This serves as a macroscopic manifestation of ZA phonon overdamping.

(iii) In Fig. 4c, the lifetime τ_{exp} is *not* consistent with the τ_{AIMD} , but being orders of magnitude lower. Also, the τ_{exp} of some low-energy phonons falls below the IR limit. This is another clear proof that the harmonic dynamics cannot capture the behavior of the phonons in InSe, proving their strongly anharmonic, overdamped nature.

In short, the AIMD calculation is based on the harmonic approximation, which cannot capture some subtle features of the phonons in experiments. The discrepancy is

not a weakness of our analysis but rather a physical feature or proof that the dynamics in InSe crystals are not harmonic but rather strongly anharmonic, leading to overdamped phonons.

(iv) By contrast, the temperature-dependent softening of the LEB phonon energies (Fig. 3a and 3b) and the Raman spectroscopies (Fig. S4), the AIMD results and our experiments show general agreements with each other.

Regarding this issue, we have added “The yellow solid line represents the Ioffe Regel limit, while the orange one is guidance for the tendency of calculating lifetime.” in the caption of Fig. 4c. We have also emphasized that the discrepancy between the calculations and the experiments is a sign of the deviation from harmonic dynamics, implying a strongly anharmonic nature of phonons.

Comment 4: In their revision, the authors have chosen to focus on the use of AIMD and, for example, simulations at much higher temperatures are employed to calculate the heat capacity all the way down to a few K. I do not understand this choice, yet it also appears to be a key point to support the existence of overdamped modes; see Line 180. More evidence for this overdamped mode will be provided in the analysis of the heat capacity. One would think, for example, that a harmonic calculation at zero temperature should be the natural starting point to compare these heat-capacity data with the calculations, as well as to establish whether it is necessary to invoke departures from this approximation at these very-low temperatures. On the experimental front, inelastic neutron-scattering data at these low temperatures would also have been the best way to compare reality (experiment) with computational predictions. These data are not reported in this work.

Response: We thank Reviewer 1 for the suggestion. We want to compare the calculation and experiments at the same (at least similar) temperatures to reach reasonable conclusions. DFT is an harmonic calculation method, which describes the structures and phonons at 0 K. However, there is no phonon excitations at 0 K and it is hardly measure the phonon spectra at a few Kelvins due to the weak signals from the thermal effect. Therefore, to understand the phonons at non-zero temperatures, we must turn

from DFT to AIMD because the latter can well include the effect of temperatures. This also follows the suggestions by the other reviewer in the last round. Here below we illustrate the rationality from the perspective of both experiment and calculation:

On the experimental front, the inelastic neutron scattering data at 50 K and 200 K are shown in Fig. R1 and give qualitatively similar results. This is reasonable since there is no structural transition for β -InSe below 300 K. Hence, we don't expect that the phonon dispersions to behave very differently at low and high temperatures. Since the data quality (intensity and statistics) is better for 200 K than 50 K, we chose to use the 200 K-data in the main text and include the 50 K-data in SI. Based on the above considerations, we prefer to use the data of 200 K at the current stage and leave neutron experiments at even lower temperature for future exploration.

Figure R1. The comparison of the inelastic neutron scattering of β -InSe at 50 K (a, b) and 200 K (c, d), respectively.

Regarding the calculation, as argued above, we turned to the AIMD since we can thus directly compare the neutron measurement with calculations at certain, non-zero temperatures. To be more specific, the AIMD simulations can include the temperature effect on the phonon dispersions, such as the broadening and softening, which cannot

be reflected by the DFT calculation. As mentioned in our reply to the General comments and Comment 3c, the AIMD simulation and experimental data are generally consistent in some phonon properties (e.g. dispersion and temperature-induced softening) to confirm the validation of the AIMD; while the differences in some subtle points (e.g. the missing ZA, lifetime) provide strong evidences for the overdamped phonons.

As a humble reminder, both the AIMD and the DFT calculation can well model the C_p data and illustrate the overdamped nature of the ZA (Fig. R2, done in the last round of reply). More precisely, both computations show that the experimental data agree with the simulations only after removing the contribution of the ZA mode.

Figure R2. The comparison of C_p/T^3 based on 200 K AIMD and zero-temperature DFT results.

Comment 5: In a similar vein, the revision falls quite short at explaining how a number of observables (particularly excitation lifetimes) have been calculated, including the level of theory and the approximations involved.

Response: Basically, the phonon lifetime τ_λ^0 is calculated by adopting the anharmonic lattice dynamics based on third-order interatomic force constant, as indicated in the Ref. [69] of the revised manuscript. Specifically, it is calculated using the single-mode relaxation time approximation (SMRTA) as implemented in ShengBTE:

$$\frac{1}{\tau_{\lambda}^0} = \frac{1}{N} \left(\sum_{\lambda'\lambda''}^{(+)} \Gamma_{\lambda\lambda'\lambda''}^{(+)} + \sum_{\lambda'\lambda''}^{(-)} \frac{1}{2} \Gamma_{\lambda\lambda'\lambda''}^{(-)} \right) + \frac{1}{N} \sum_{\lambda'}^{(iso)} \Gamma_{\lambda\lambda'}^{(iso)} \quad (1)$$

where N is the total number of \mathbf{q} -points in the first Brillouin zone, λ denotes a phonon mode characterized by its wave vector \mathbf{q} and branch. The superscript (\pm) indicates the three-phonon processes. The scattering rates Γ for different processes are calculated through the scattering probability matrix:

$$\begin{aligned} \Gamma_{\lambda\lambda'\lambda''}^{(+)} &= \frac{\hbar\pi}{4} \frac{n_{\lambda'}^0 - n_{\lambda''}^0}{\omega_{\lambda}\omega_{\lambda'}\omega_{\lambda''}} |V_{\lambda\lambda'\lambda''}^{(+)}|^2 \delta(\omega_{\lambda} + \omega_{\lambda'} - \omega_{\lambda''}) \\ \Gamma_{\lambda\lambda'\lambda''}^{(-)} &= \frac{\hbar\pi}{4} \frac{n_{\lambda'}^0 + n_{\lambda''}^0 + 1}{\omega_{\lambda}\omega_{\lambda'}\omega_{\lambda''}} |V_{\lambda\lambda'\lambda''}^{(-)}|^2 \delta(\omega_{\lambda} - \omega_{\lambda'} - \omega_{\lambda''}) \end{aligned} \quad (2)$$

where n_{λ}^0 and ω_{λ} are the Bose-Einstein distribution at equilibrium and the frequency for a certain mode λ , respectively. Conservation of energy is enforced by the Dirac delta function δ . The matrix elements V are given by the Fourier transformation of force constants, or transition probability matrices:

$$V_{\lambda\lambda'\lambda''}^{(\pm)} = \sum_{ijk} \sum_{\alpha\beta\gamma} \Phi_{ijk}^{\alpha\beta\gamma} \frac{e_{\alpha}^{\lambda}(i) e_{\beta}^{\pm\lambda'}(j) e_{\gamma}^{-\lambda''}(k)}{\sqrt{M_i M_j M_k}} e^{\pm i\mathbf{q}' \cdot \mathbf{r}_j} e^{-i\mathbf{q}'' \cdot \mathbf{r}_k} \quad (3)$$

where i, j, k denote the atomic indices and α, β, γ denote the Cartesian dimensions x, y or z . $\Phi_{ijk}^{\alpha\beta\gamma}$ are the third-order interatomic force constants. $e_{\alpha}^{\lambda}(i)$ is the eigenvector component for a phonon mode. \mathbf{r}_j is the position vector of the unit cell where j th atom lies, and M_j is its mass. Therefore, based on the above equations, after fitting the second-order and third-order interatomic based on AIMD simulations, one can obtain the phonon dispersion and phonon lifetimes correspondingly. **Some adjustment of the text has been made in the revised manuscript.**

Reviewer #2

General comments: The authors made a good effort to revise the manuscript extensively. They address most of my concerns and I believe the manuscript is suitable for publication in the current form.

Response: We appreciate the reviewer's favorable assessment of this manuscript.

Reviewer #3

General comments: The authors have revised their work as requested by the different reviewers, which has very much improved their article. Their work can now be accepted for publication in Nature Communication after correcting small typo error indicated below. There is still a small typo error in line 206: in ref. 21, the compound studied is AgSbTe₂ and not AgSnTe₂ as written in the revised manuscript.

Response: We appreciate the reviewer's favorable assessment of this manuscript and have corrected this typo in the revised manuscript.

REVIEWERS' COMMENTS

Reviewer #1 (Remarks to the Author):

-

Reviewer #2 (Remarks to the Author):

The authors sufficiently addressed address some of Reviewer 1's comments in the revised manuscript. This includes rewriting the statement about "anomalous" heat capacity (labelled as Comment 2 in the rebuttal) and how phonon life was calculated (Comment 5).

Many of Reviewer 1's other comments (Comment 1 and 3) are around the concern about the observation of "overdamped" phonons. While the effect is weak in the slices and constant-q cuts, it is there. It should be noted that the effect shows up in the 4D neutron scattering data from TOF spectrometer and a relatively low single-to-noise ratio does not undermine that observation. As for Comment 4, while applying phonon DOS obtain from AIMD at 200K to calculation heat capacity misses the temperature dependent phonon renormalization, it is expected this shall not change the conclusion qualitatively.

In summary, I believe the manuscript is suitable for publication. The author might consider by including a version of Fig. 2 without the over-plotted dispersion calculation in SI for better demonstration of the overdamped mode.

Reviewer #3 (Remarks to the Author):

In the present revised version , the authors have made changes for satisfying the comments of reviewer 1 whereas the two requests of the two other reviewers have been statisfied, helping to improve the mansucript.

The main concern of the reviewer 1 is the claim of unambiguous low-energy damped phonons. The author's reply partly answered this concern but I think that the authors must take more cautious

and add a part of their answer in the article to make clearer their argument in favor to the overdamped phonons.

In view of that, I request :

- to change the title " Overdamped Phonons: Uncovering the Consequences of Interlayer Slip in 2D van der 2 Waals InSe Crystals "

into " Possible overdamped Phonons: Uncovering the Consequences of Interlayer Slip in 2D van der

2 Waals InSe Crystals "

- to be less affirmative along the text and take more caution when writing about overdamped phonons. There are indeed several indications of possible overdamped phonons but not more.
- to clearly write in the text the four main reasons given in the author's reply. However, the authors must take caution with the analysis of the heat capacity data as this is just the analysis of the deviation between the experimental data and calculated data with removing the contribution of ZA branche is consistent with the previous observation. They must to avoid too strong statement concerning the Cp analysis because there is often some deviation between experiments and DFT at low temperature notably because of any small difference is amplified by the Debye plot, even with only few % of difference between experiments and DFT calculations.
- that the authors must also use only the term "deviation from the Debye behaviour" instead of term "glassy behavior" which would wrongly involved that deviation from Debye behaviour happens only or mainly in glasses, which is obviously wrong.
- to clearly write that the intensity of the calculated phonon dispersion in figures 2c-d have been obtained with the calculated dynamical structure factor from AIMD and with linewidth of the instrumental resolution.
- to show the Cp calculated with standard harmonic calculations in the figure S8 in order to show that it gives almost the same results than the AIMD calculations.

If the authors follow well the above requests, I think that the next revised version of the present work will be improved and could be accepted for publication in Nature Communications.

Reviewer 2:

General comments: The authors sufficiently address some of Reviewer 1's comments in the revised manuscript. This includes rewriting the statement about heat capacity (labelled as Comment 2 in the rebuttal) and how phonon life was calculated (Comment 5).

Many of Reviewer 1's other comments (Comment 1 and 3) are around the concern about the observation of overdamped phonons. While the effect is weak in the slices and constant-q cuts, it is there. It should be noted that the effect shows up in the 4D neutron scattering data from TOF spectrometer and a relatively low signal-to-noise ratio does not undermine that observation.

As for Comment 4, while applying phonon DOS obtained from AIMD at 200K to calculate heat capacity misses the temperature-dependent phonon renormalization, it is expected this shall not change the conclusion qualitatively.

In summary, I believe the manuscript is suitable for publication. The author might consider including a version of Fig. 2 without the over-plotted dispersion calculation in SI for better demonstration of the overdamped mode.

Response: We are grateful to the Reviewer for the recognition of our research. Following the suggestion, we have included a version of the Fig. 2(a-b) without the over-plotted dispersion calculation in SI of the revised manuscript (Figure R1, also Fig. S4).

Figure R1. The phonon dispersions observed in the dynamical susceptibility $\chi''(Q, E)$ from (a),(b) inelastic neutron scattering (INS) measurements. (c) The phonon spectra

of whole energy scale along [K-K0] at 50 K. (d), (e) Temperature dependent phonon dispersion by AIMD simulations (with and without LO-TO splitting). (f) Temperature-dependence of Raman spectra experimentally.

Reviewer 3:

General comments: In the present revised version, the authors have made changes for satisfying the comments of reviewer 1 whereas the two requests of the two other reviewers have been satisfied, helping to improve the manuscript. The main concern of the reviewer 1 is the claim of unambiguous low-energy damped phonons.

The author's reply partly answered this concern but I think that the authors must take more caution and add a part of their answer in the article to make clearer their argument in favor to the overdamped phonons.

Response: We greatly appreciate the reviewer's positive feedback and constructive comments on this manuscript.

In view of that, I request :

1. to change the title " Overdamped Phonons: Uncovering the Consequences of Interlayer Slip in 2D van der Waals InSe Crystals " into " Possible overdamped Phonons: Uncovering the Consequences of Interlayer Slip in 2D van der Waals InSe Crystals "

Response: Following the comments and the editorial suggestion, we have changed the title into "Uncovering the phonon spectra and lattice dynamics of plastically deformable InSe van der Waals crystals" to tune down the tone.

2. to be less affirmative along the text and take more caution when writing about overdamped phonons. There are indeed several indications of possible overdamped phonons but not more.

Response: We appreciate the the suggestion of the reviewer and the editors regarding this issue. In the revised manuscript, we have made adjustments to the main text and abstract to be less affirmative, incorporating phrases like "possible" and "might be" to soften our statements.

3. to clearly write in the text the four main reasons given in the author's reply. However, the authors must take caution with the analysis of the heat capacity data as this is just the analysis of the deviation between the experimental data and calculated data with removing the contribution of ZA branche is consistent with the previous observation. They must to avoid too strong statement concerning the C_p analysis because there is often some deviation between experiments and DFT at low temperature notably because of any small difference is amplified by the Debye plot, even with only few % of difference between experiments and DFT calculations.

Response: We thank the reviewer for this suggestion. We have added the four main reasons supporting the overdamped-like nature of the phonons in the conclusion section. Moreover, we have revised the discussion on C_p to avoid very strong statements.

4: that the authors must also use only the term "deviation from the Debye behaviour" instead of term "glassy behavior" which would wrongly involved that deviation from Debye behaviour happens only or mainly in glasses, which is obviously wrong.

Response: The term "glassy behavior" has been changed into "deviation from the Debye behaviour" when discussing about the specific heat in the manuscript.

5: to clearly write that the intensity of the calculated phonon dispersion in figures 2c-d have been obtained with the calculated dynamical structure factor from AIMD and with linewidth of the instrumental resolution.

Response: We have included the sentence "Fig. 2c-d are intensity plots considering the calculated dynamical structure factor from AIMD and instrument resolution" in both the caption of Figure 2 and the Methods section.

6: to show the C_p calculated with standard harmonic calculations in the figure S8 in order to show that it gives almost the same results than the AIMD calculations.

Response: We have included the C_p calculated with standard harmonic calculations in Figure R2 below, also Fig. S8, showing that the two calculation methods give similar results.

Figure R2. (a) The calculated mode-contributed C_p/T^3 for the three acoustic modes (ZA, TA and LA) and for all optical modes; (b) The C_p/T^3 data include curves from both AIMD and first-principles simulations (DFT, standard harmonic calculation at 0 K): the light purple and red curves are generated from AIMD, with the light purple considering all phonon DOS and the red excluding the ZA contribution. The blue and green dashed curves are from first-principles simulations, with the blue dashed line considering all phonon DOS and the green dashed line excluding the ZA contribution.